# Understanding Multi-Granularity
# for Open-Vocabulary Part Segmentation

**Jiho Choi**[1][*], **Seonho Lee**[1][*], **Seungho Lee**[2], **Minhyun Lee**[2], **Hyunjung Shim**[1][†]

[1]Graduate School of Artificial Intelligence, KAIST, Republic of Korea
[2]School of Integrated Technology, Yonsei University, Republic of Korea
{jihochoi, glanceyes, kateshim}@kaist.ac.kr, {seungholee, lmh315}@yonsei.ac.kr

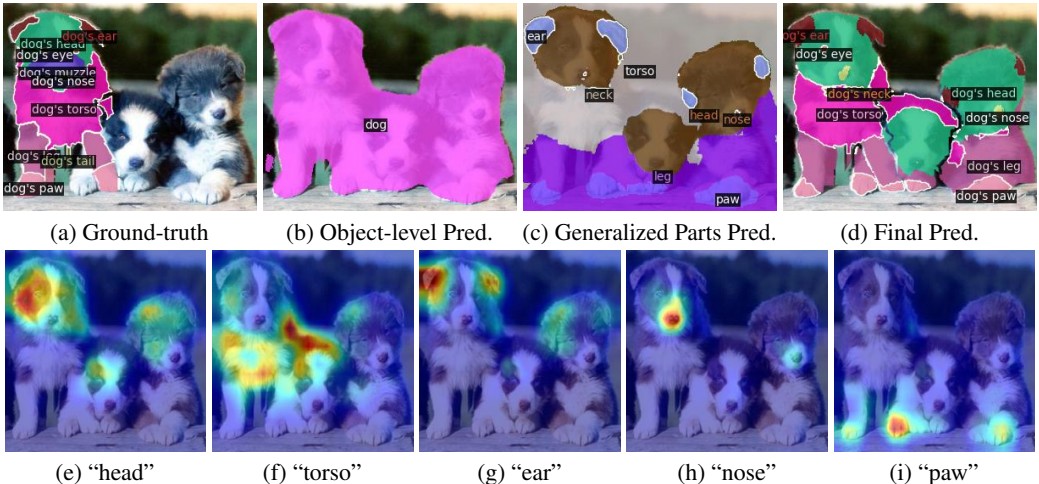

|  |  |  |  |
|---|---|---|---|
| (a) Ground-truth | (b) Object-level Pred. | (c) Generalized Parts Pred. | (d) Final Pred. |

|  |  |  |  |  |
|---|---|---|---|---|
| (e) "head" | (f) "torso" | (g) "ear" | (h) "nose" | (i) "paw" |

Figure 1: Prediction results of our PartCLIPSeg for unseen categories in the Pascal-Part-116 [7, 46] validation set. A "dog" is **unseen** during training. The final prediction of PartCLIPSeg utilizes (b) object-level context and (c) generalized parts, incorporating disjoint activation among (e)–(i) parts, and enhancing activation for smaller parts (e.g., (h) "nose").

## Abstract

Open-vocabulary part segmentation (OVPS) is an emerging research area focused on segmenting fine-grained entities using diverse and previously unseen vocabularies. Our study highlights the inherent complexities of part segmentation due to intricate boundaries and diverse granularity, reflecting the knowledge-based nature of part identification. To address these challenges, we propose PartCLIPSeg, a novel framework utilizing generalized parts and object-level contexts to mitigate the lack of generalization in fine-grained parts. PartCLIPSeg integrates competitive part relationships and attention control, alleviating ambiguous boundaries and underrepresented parts. Experimental results demonstrate that PartCLIPSeg outperforms existing state-of-the-art OVPS methods, offering refined segmentation and an advanced understanding of part relationships within images. Through extensive experiments, our model demonstrated a significant improvement over the state-of-the-art models on the Pascal-Part-116, ADE20K-Part-234, and PartImageNet datasets. Our code is available at https://github.com/kaist-cvml/part-clipseg.

---

[*]Equal contribution
[†]Corresponding author

38th Conference on Neural Information Processing Systems (NeurIPS 2024).

# 1 Introduction

The pursuit of understanding parts and multi-granularity in computer vision [7, 13, 21] mirrors the innate complexities of animal instincts. For example, a "cheetah" instinctively targets an "impala's neck" during a hunt, demonstrating its ability to distinguish specific parts. This ability extends to applications such as robot commands [44], fine-grained controls on image editing [31], and more sophisticated image generation [45]. Part segmentation aims to mimic this ability by recognizing intricate details (e.g., parts) within objects, going beyond simple object-level segmentation to achieve detailed and diverse entity recognition.

Recognizing parts is more challenging than recognizing whole objects due to their complexity and diversity. Parts often have ambiguous boundaries not only defined by visual cues but also require a broader spectrum of contextual information, reflecting their knowledge-based nature. For example, the "head" of a "dog" may include only the "face" or also the "neck" depending on the annotators' perspective [7, 21].

To address difficulties in part segmentation, Open-Vocabulary Part Segmentation (OVPS) [40, 44, 46] has evolved by leveraging the knowledge of powerful Vision-Language Models (VLMs) like CLIP [38] or ALIGN [24]. Especially, it aims to achieve adaptive recognition and processing of previously unseen categories with the aid of pre-trained VLMs, pushing the boundaries of vocabularies in traditional part segmentation. By utilizing Oracle supervision of base classes during training, recent studies in OVPS exploit part-level knowledge of base classes to generalize to novel classes. Recently, VLPart [40] uses DINO [5] features to map correspondences between base and novel classes and creates pseudo labels for the novel categories. OV-PARTS [46] addresses the ambiguity of part boundaries by introducing object mask prompts and transferring knowledge of base class through a few-shot approach. These methods successfully extract knowledge from VLMs and extend it to novel classes, achieving significant performance improvements in open-vocabulary settings.

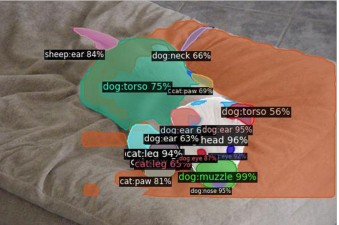 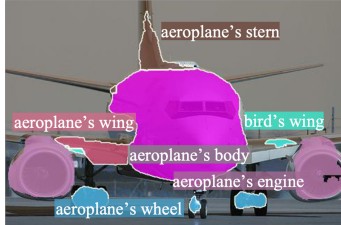 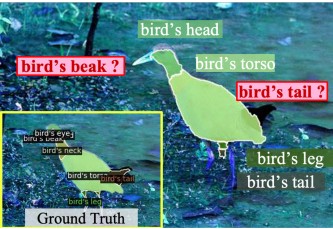

(a) Lack of generalization      (b) Ambiguous boundaries      (c) Missing underrepresented part

Figure 2: Limitations of existing OVPS methods in predicting unseen categories. (a) Lack of generalization: Classification of a "dog's parts" involving categories like "cats" and "sheep", "dog's tail" misclassified as "sheep's ear". (VLPart [40]) (b) Ambiguous boundaries: Vague boundary output of "aeroplane's body". (c) Missing underrepresented parts: Neglecting parts such as "beak" and "leg". (CLIPSeg [32, 46]).

However, through empirical analysis of existing OVPS methods, we observed several common limitations in Figure 2. (Lack of generalization in (a)) Despite understanding part-level information, they often misidentify parts at the object level, e.g., a "dog's leg" as a "cat's leg". Also, part-level misclassification occurs as the knowledge of parts in the base class fails to generalize to a novel class, e.g., "dog's tail" as a "sheep's ear". (Ambiguous boundaries of parts in (b)) They fail to maintain non-overlapping relationships between parts, frequently resulting in overlaps, e.g., an "airplane's wing" overlapping with its "body" or the presence of empty spaces where no part is predicted. (Missing underrepresented parts in (c)) They ignore small and less frequent parts, causing prediction bias based on part size.

To overcome these limitations, we propose a novel framework called PartCLIPSeg, which consists of three main components. First, we devise generalized parts with object-level contexts to address the lack of generalization issue as the upper side of Figure 1. It explicitly obtains object-level and part-level pseudo-labels from VLMs and trains the OVPS model to satisfy both types of supervision. This guides the model to learn object boundaries while recognizing both part and object-level classes. Then, we suggest an attention control for minimizing the overlap between predicted parts, ensuring that parts are clearly separated as the lower side of Figure 1. In this way, we effectively leverage

internal part information to learn ambiguous part boundaries. Finally, we enhance the activation related to certain parts by normalizing the activation scale of CLIP's self-attention information. It prevents small and less frequent areas from being ignored in pseudo-labels. This strategy ensures that the smallest granularity levels are retained in the final prediction. Through these three modules, PartCLIPSeg effectively addresses the challenges of existing OVPS methods and achieves robust multi-granularity segmentation. As a result, the proposed method achieves significant improvements in mIoU for both unseen and the harmonic mean when compared to previous state-of-the-art methods on Pascal-Part-116, ADE20K-Part-234, and PartImageNet in both Pred-All and Oracle-Obj settings.

## 2    Related Work

**Open-Vocabulary Semantic Segmentation.** Open-vocabulary [19, 55] semantic segmentation (OVSS) goes beyond traditional semantic segmentation, which is restricted to predefined categories, by enabling predictions for unseen classes. Pioneering works focused on aligning predefined text embeddings with pixel-level visual features [4, 48, 56]. By leveraging large-scale Vision-Language Models (VLMs) like CLIP [38] and ALIGN [24], OVSS enables zero-shot segmentation through rich multi-modal features learned from extensive image-text pairs. MaskCLIP [58] modified CLIP's image encoder to directly handle visual and text features for segmenting novel classes. Some works proposed two-stage strategy [15, 16, 18, 20, 29, 30, 51, 52]: first, models generate class-agnostic mask proposals [9, 10]; then, a pre-trained VLM predicts the category for each region. Some studies have introduced diffusion models to improve mask generation quality [51] or fine-tuned CLIP to enhance classification capabilities [20, 29]. Other studies have adopted a single-stage framework [11, 27, 32, 49, 54, 59]. They use pre-trained CLIP models to align pixel-level visual features with text features. CLIPSeg [32] adds a transformer-based pixel decoder with a FiLM [17] module to fuse multi-modal features. ZegCLIP [59] enhances segmentation by incorporating learnable tokens. SAN [53] adopted a side adapter network for a CLIP-aware end-to-end approach to predict proposal-wise classification. FC-CLIP [54] uses a frozen convolutional CLIP to predict class-agnostic masks and classifies using mask-pooled features [54]. CAT-Seg [11] and SED [49] generate pixel-level cost maps and refine them for segmentation.

**Part Segmentation.** Part segmentation aims to identify the individual parts of objects, a task that is more complex and costly due to the smaller and more diverse nature of parts compared to whole objects. To tackle this, various datasets like Pascal-Part [7], PartImageNet [21], ADE20k-Part [57], Cityscapes-Panoptic-Part [13], and PACO [2] provide diverse and detailed part annotations. Earlier studies [7, 12, 22, 23, 43] used self-supervised constraints and contrastive settings for effective part-level entity segmentation. Recent studies extended this to open-vocabulary scenarios [35, 40, 46], opening new avenues for handling diverse parts. By leveraging class-agnostic detectors [35] and Vision-Language Models like CLIP [40, 46], part segmentation has extended its generalization ability to unseen parts. Our work builds upon and extends methodologies from these studies.

## 3    Methodology

As illustrated in Figure 2, we identified three primary challenges of open-vocabulary part segmentation (OVPS): lack of generalization, overlapping parts, and missing underrepresented parts. Recognizing object-specific parts (such as "dog's torso") cannot be determined solely by looking at each part in isolation; it is imperative to consider both generalized part information and the overall context of the object. Furthermore, some parts may have overlapping meanings across different granularity labels (e.g., "eye", "face", and "head"). This implies that predictions should consider direct guidance for each part as well as the relationships between different parts. These intricate spatial and functional dependencies between parts are crucial for achieving a holistic understanding and precise predictions in fine-grained entity segmentation tasks.

Based on this motivation, we propose a novel OVPS method, `PartCLIPSeg`. This method leverages *generalized part* information combined with *object-level context* to tackle the lack of generalization problem (see Section 3.2). Also, we directly minimize the overlap among part predictions to improve the part boundaries (see Section 3.3.1). Finally, we normalize the scale of attention activation from various parts for handling missing underrepresented parts (see Section 3.3.2). The overall architecture of our method is shown in Figure 3.

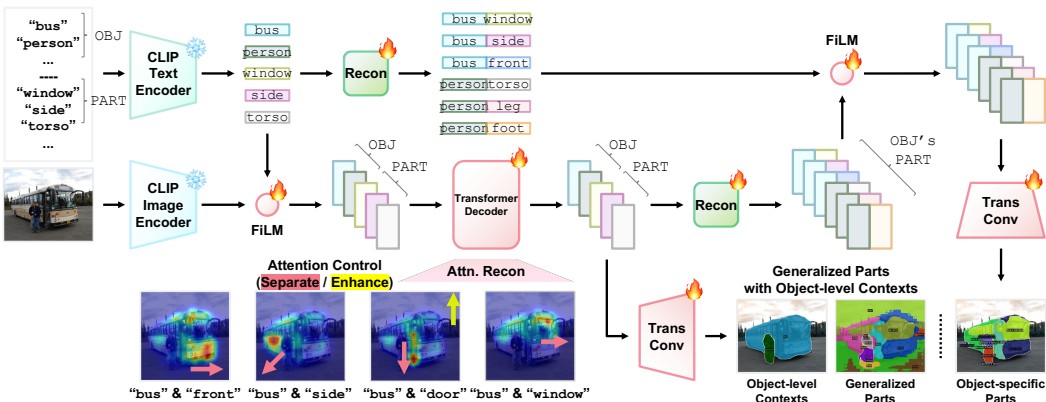

Figure 3: **The overall architecture of PartCLIPSeg.** The embeddings derived from the object category name and the part category name are conditioned using the FiLM operation. Each embedding, modified through attention control, is subsequently reconstructed to predict the final object-specific part results.

## 3.1 Preliminary

OVPS aims to segment an image into a set of `object-specific part` categories $\mathbf{C}^{\text{test}}_{\text{obj-part}}$ (e.g., "dog's head," "car's front") in the *test* set, where the image is $\mathcal{I} \in \mathbb{R}^{H \times W \times 3}$, and $H$ and $W$ are the height and width. During training, image-mask pairs $\{(\mathcal{I}_k, \mathcal{M}_k)\}$ are used, consisting of images $\mathcal{I}_k$ and corresponding ground-truth mask $\mathcal{M}_k$ which only contains the `object-specific part` categories $\mathbf{C}^{\text{train}}_{\text{obj-part}}$ (e.g., "cat's head," "bus's front") in the *train* set.

**Zero-Shot Part Segmentation.** Open-vocabulary is a generalized zero-shot task, allowing the zero-shot segmentation protocol to evaluate zero-shot part segmentation performance. In this setting, *train* and *test* category names are divided into *seen* (base) and *unseen* (novel) sets, respectively, with disjoint object-specific category names; $\left\{ \mathbf{C}^{\text{unseen}}_{\text{obj-part}} \cap \mathbf{C}^{\text{seen}}_{\text{obj-part}} = \varnothing \right\}$.

**Cross-Dataset Part Segmentation.** In this setting, the model is trained on one dataset and evaluated on another without fine-tuning. This means that the category names of the *train* and *test* sets come from different datasets, denoted as $\mathbf{C}^{\text{train}}_{\text{obj-part}} \neq \mathbf{C}^{\text{test}}_{\text{obj-part}}$. Considering the domain gap between the datasets, such as differences in granularity, this setting is more challenging.

## 3.2 Generalized Parts with Object-level Contexts

To address the problem of a lack of generalization, we propose leveraging generalized parts with object-level contexts. The concept of generalized parts involves identifying and utilizing common structural components that are shared across different object-level categories. For instance, many animals have parts like "head" or "torso" which, although functionally and visually distinct, may share certain underlying characteristics. By introducing generalized parts from object-specific parts, our PartCLIPSeg can efficiently recognize and segment these object-specific parts across diverse object classes, significantly enhancing the model's ability to generalize from seen to unseen categories.

Although generalized parts help distinguish the part-level categories, the visual information of a part may not suffice for accurately classifying their object-level categories. For instance, predicting the "leg" part of an animal can be challenging to identify when solely examining the part as it may not clearly indicate to which animal it belongs. For this reason, there have been attempts to incorporate object-level guidance [33, 40, 46] in part segmentation. However, object-level guidance without a generalized part may lose contextual information and miss hierarchical relationships.

By integrating object contexts with generalized parts, PartCLIPSeg employs object-level guidance that captures the holistic essence of the object to which parts belong. This integration allows for a more precise understanding and classification of parts, improving the overall performance of OVPS.

**Object and Part Embedding Generation.** We modified the architecture of CLIPSeg [32, 46], which adopted CLIP [38] encoder-decoder architecture for semantic segmentation. However, it is worth

noting that our approach of utilizing generalized parts with object-level context is orthogonal to other previously proposed object-level segmentation methods [5, 26, 28, 39].

The proposed approach begins by parsing an object-specific part category name, $\mathbf{c}_{\text{obj-part}} \in \mathbf{C}_{\text{obj-part}}$, into separate components: an object category name ($\mathbf{c}_{\text{obj}}$) and a generalized part category name ($\mathbf{c}_{\text{part}}$), e.g., "cat" and "torso". Then, the CLIP text encoder, $\text{CLIP}_{\mathcal{T}}^*(\cdot)$, is used to transform these category names into their respective CLIP embeddings ($\mathbf{e}_{\text{obj}}^{\mathcal{T}}$ and $\mathbf{e}_{\text{part}}^{\mathcal{T}}$). It will condition the image features, $\mathbf{e}^{\mathcal{I}}$, derived from the CLIP image encoder, $\text{CLIP}_{\mathcal{I}}^*(\cdot)$ as:

$$\mathbf{e}_{[\text{obj} \mid \text{part}]}^{\mathcal{T}} = \text{CLIP}_{\mathcal{T}}^*(\mathbf{c}_{[\text{obj} \mid \text{part}]}), \mathbf{e}^{\mathcal{I}} = \text{CLIP}_{\mathcal{I}}^*(\mathcal{I}), \tag{1}$$

where $*$ denotes frozen pre-trained models. By using Feature-wise Linear Modulation (FiLM) [36, 42], each category name embeddings respectively modulate the image features as:

$$\mathbf{e}_{[\text{obj} \mid \text{part}]}^{\mathcal{I}} = \mathbf{e}^{\mathcal{I}} \oplus \text{FiLM}(\mathbf{e}_{[\text{obj} \mid \text{part}]}^{\mathcal{T}}), \tag{2}$$

where $\oplus$ is an element-wise sum. FiLM is an adaptive affine transformation widely used for multi-modal or conditional tasks. It helps retrieve adequate conditioning for the image features. The modulated image features, $\mathbf{e}_{[\text{obj} \mid \text{part}]}^{\mathcal{I}}$, corresponding to each object and part category name, pass through a decoder module. The decoder module will be discussed in detail in Section 3.3. They then proceed through a transposed convolution model. Finally, the output mask of the object $\hat{s}^o$ and part $\hat{s}^p$ are evaluated with ground-truth mask of objects, $s^o$, and parts, $s^p$. Oracle supervision for the object and parts mask is simply computed from a combination of object-specific parts annotations: $s \in \mathcal{M}$.

**Object-specific Part Construction.** We utilize previously computed generalized part embeddings ($\mathbf{e}_{\text{part}}^{\mathcal{I}}$, $\mathbf{e}_{\text{part}}^{\mathcal{T}}$) and object embeddings ($\mathbf{e}_{\text{obj}}^{\mathcal{I}}$, $\mathbf{e}_{\text{obj}}^{\mathcal{T}}$) to reconstruct object-specific part embeddings. This process involves separate operations on modulated image features and category name embeddings.

Initially, we project the concatenated results of the object category name with the generalized part category name. This is to synthesize the embeddings for the target object-specific part category name. The approach ensures that the resultant embeddings are highly representative of parts and contextually relevant. The equivalent operation is applied to both object-level image features and part-level image features to generate object-specific image features as:

$$\mathbf{e}_{\text{obj-part}}^{[\mathcal{T}|\mathcal{I}]} = \text{Proj}\left(\left[\mathbf{e}_{\text{obj}}^{[\mathcal{T}|\mathcal{I}]} \mid \mathbf{e}_{\text{part}}^{[\mathcal{T}|\mathcal{I}]}\right]\right). \tag{3}$$

The resulting object-specific part embeddings are further refined by a FiLM process. Combined with the respective object-specific image features, final modulated object-specific part embeddings, $\mathbf{e}_{\text{obj-part}}$ is computed as:

$$\mathbf{e}_{\text{obj-part}} = \mathbf{e}_{\text{obj-part}}^{\mathcal{I}} \oplus \text{FiLM}(\mathbf{e}_{\text{obj-part}}^{\mathcal{T}}). \tag{4}$$

These embeddings are then processed through a deconvolution layer to produce the final segmentation masks $s \in \mathcal{M}$. This step ensures that the embeddings are precisely aligned to enhance the definition and accuracy of the object-specific part masks. It effectively bridges the gap between object and part-level categorical information with object-specific parts information.

**Object, Part, and Object-specific Part Mask Supervision.** The mask supervision is provided for three distinct categories: object-specific parts, objects, and generalized parts. This multi-faceted supervision enables our model to effectively disentangle generalized parts from objects, thereby facilitating a more nuanced learning process for OVPS. This disentanglement is crucial for the model to accurately recognize and differentiate between various object categories and their corresponding parts. It enhances the model's ability to handle complex segmentation tasks with unseen object-specific parts. The overall mask guidance loss can be defined as follows:

$$\mathcal{L}_{\text{mask}} = \sum_{i=1}^{|\mathbf{C}_{\text{obj-part}}|+1} \underbrace{\text{BCE}(s_i, \hat{s}_i)}_{\text{object-specific part}} + \lambda_{\text{obj}} \sum_{i=1}^{|\mathbf{C}_{\text{obj}}|+1} \underbrace{\text{BCE}(s_i^o, \hat{s}_i^o)}_{\text{object guidance}} + \lambda_{\text{part}} \sum_{i=1}^{|\mathbf{C}_{\text{part}}|} \underbrace{\text{BCE}(s_i^p, \hat{s}_i^p)}_{\text{generalized part guidance}}, \tag{5}$$

where $|\mathbf{C}_{\text{obj-part}}| + 1$ and $|\mathbf{C}_{\text{obj}}| + 1$ are for uncategory (or background) prediction. The disentangled object and part generalization with object-specific parts guidance provides a clue to the lack of generalization problem.

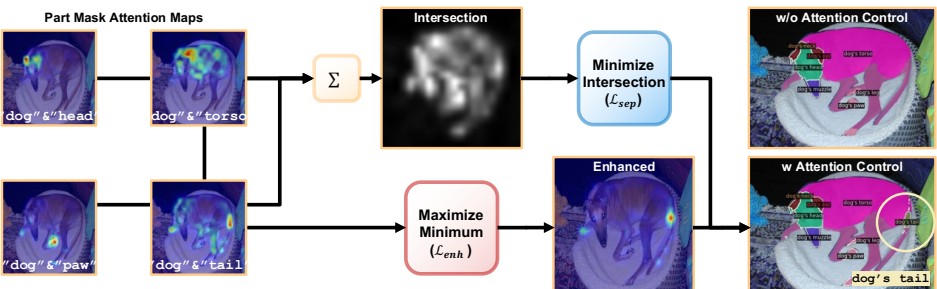

Figure 4: Example of attention control using separation and enhance losses. The proposed method manipulates attention maps to accurately identify and segment small parts.

### 3.3 Attention Control for Ambiguity and Omission

In this subsection, we address the previously mentioned challenges: (1) ambiguity in part boundaries and (2) omission of small or infrequently appearing parts. The main reason for these challenges is the incomplete guidance from knowledge-based, multi-granularity characteristics of parts. To overcome these, we adopt unsupervised methods traditionally used in fine-grained recognition and part discovery studies [7, 12, 43]. Specifically, we utilize approaches for adjusting self-attention activation inspired by the recent diffusion methods [6, 25, 41].

We assume that the distribution of self-attention activation maps for visual tokens belonging to the same object-specific part mask should exhibit inter-similarity characteristics [41], implying similar distributions. To this end, we first compute the average self-attention map $\mathcal{A}_{\mathcal{M}_\mathbf{c}}$ for each object-specific part mask $\mathcal{M}_\mathbf{c}$, where $\mathbf{c} \in \mathbf{C}_{\text{obj-part}}$ represents an object-specific part category. This is done by summing the self-attention activation maps from channels specifically corresponding to object $\mathbf{c}_{\text{obj}}$ and part $\mathbf{c}_{\text{part}}$, across all spatial tokens $(h, w)$ within the mask, as follows:

$$\mathcal{A}_{\mathcal{M}_\mathbf{c}} = \frac{1}{|\mathcal{M}_\mathbf{c}|} \sum_{(h,w) \in \mathcal{M}_\mathbf{c}} \left( \mathcal{A}_{\mathbf{c}_{\text{obj}}}[h, w, :, :] + \mathcal{A}_{\mathbf{c}_{\text{part}}}[h, w, :, :] \right). \tag{6}$$

Subsequently, the self-attention map $\mathcal{A}_{\mathcal{M}_\mathbf{c}}$ for the object-specific part mask is refined through min-max normalization, followed by the application of a Gaussian filter to smooth the initial activation as in [6, 50]. Therefore, the dimensions of both the original and normalized self-attention maps for the object-specific part masks are as follows: $\mathcal{A}_{\mathcal{M}_\mathbf{c}}, \mathcal{A}_{\mathcal{M}_\mathbf{c}}^{\text{norm}} \in \mathbb{R}^{H \times W}$.

#### 3.3.1 Minimizing Part Overlaps for Ambiguity

In the self-attention of the decoder layers, competition between object-specific parts helps define boundaries that cannot be sufficiently established by supervision alone. Using the previously obtained normalized attention map, our method generates parts with minimized intersections, inspired by [1, 3, 25, 37, 47]. This approach effectively mitigates the ambiguity issue in part boundaries. Specifically, the normalized attention activation map $\mathcal{A}_{\mathcal{M}_\mathbf{c}}^{\text{norm}}$ is first binarized based on an arbitrary threshold $\gamma$ as:

$$\mathcal{B}_{\mathcal{M}_\mathbf{c}}(h, w) = \mathbf{1}_{\{\mathcal{A}_{\mathcal{M}_\mathbf{c}}^{\text{norm}}(h,w) \geq \gamma\}}, \tag{7}$$

where $\mathcal{B}_{\mathcal{M}_\mathbf{c}}$ denotes binarized attention map for part mask $\mathcal{M}_\mathbf{c}$. From now on, $\mathbf{C}_{\text{obj-part}}$ is simply denoted as $\mathbf{C}$. The separation loss $\mathcal{L}_{\text{sep}}$, which indicates the degree of intersection between object-specific parts, is as follows:

$$\mathcal{L}_{\text{sep}} = \frac{1}{|\mathbf{C}|} \left| \frac{\left\{ (h, w) \mid \sum_{\mathbf{c} \in \mathbf{C}} \mathcal{B}_{\mathcal{M}_\mathbf{c}}(h, w) > 1 \right\}}{\left\{ (h, w) \mid \sum_{\mathbf{c} \in \mathbf{C}} \mathcal{B}_{\mathcal{M}_\mathbf{c}}(h, w) \geq 1 \right\}} \right|, \tag{8}$$

where separating activation mitigates the challenge of ambiguous boundaries between parts.

#### 3.3.2 Enhancing Part Activation for Omission

To address the omission problem, we employ a method inspired by attention controls in modern diffusion-based approaches [3, 6]. This approach enhances the activation within the self-attention

activation map to enhance underrepresented parts before normalization. Specifically, for each object-specific part mask, the maximum value within the attention map is identified. Subsequently, among all object-specific parts, the minimum activation of the part with the maximum value is enhanced as:

$$\mathcal{L}_{\text{enh}} = 1 - \min_{\mathbf{c} \in \mathbf{C}} \left( \max_{(h,w) \in \mathcal{M}_{\mathbf{c}}} \mathcal{A}_{\mathcal{M}_{\mathbf{c}}}[h,w] \right), \tag{9}$$

thereby boosting its representational efficacy. In this way, the enhancement loss $\mathcal{L}_{\text{enh}}$ provides sufficient guidance for small or infrequently occurring parts, effectively mitigating the omission problem.

The training objective for PartCLIPSeg integrates three key loss components as:

$$\mathcal{L}_{\text{all}} = \mathcal{L}_{\text{mask}} + \lambda_{\text{sep}}\mathcal{L}_{\text{sep}} + \lambda_{\text{enh}}\mathcal{L}_{\text{enh}}, \tag{10}$$

where (1) $\mathcal{L}_{\text{mask}}$ for generalized parts with object-level context, (2) $\mathcal{L}_{\text{sep}}$ for addressing ambiguous boundaries, (3) $\mathcal{L}_{\text{enh}}$ for handling missing underrepresented parts, and $\lambda_{\text{sep}}$ and $\lambda_{\text{enh}}$ are hyperparmeters.

## 4 Experiments

### 4.1 Experimental Setups

**Datasets.** We evaluate our method on three part segmentation datasets: Pascal-Part-116 [7, 46], ADE20K-Part-234 [46, 57], and PartImageNet [21]. Pascal-Part-116 [7, 46] consists of 8,431 training images and 850 test images. It is a modified version of PascalPart [7] by removing direction indicators for certain part classes and merging them to avoid overly complex part definitions. This dataset contains a total of 116 object part classes across 17 object categories. ADE20K-Part-234 [46, 57] consists of 7,347 training images and 1,016 validation images. It provides instance-level object mask annotations along with their corresponding part mask annotations, including 44 objects and 234 parts. PartImageNet [21] contains 16k training images and 2.9k validation images, segmented into 158 object classes from ImageNet [14] and organizes them into 11 super-categories. For this study, we select 40 object classes that represent common categories to assess cross-dataset performance effectively. More details about the datasets can be found in the supplementary materials.

**Evaluation Protocols.** We use two evaluation protocols for the performance of OVPS: (1) **Pred-All** setting, where the ground truth object-level mask and object class are not provided, and (2) **Oracle-Obj** setting, where the ground truth object-level mask and object class are known. In particular, the **Pred-Obj** setting in OV-PARTS [46] uses predicted masks from the off-the-shelf segmentation model. In contrast, our **Pred-All** setting is a more challenging and practical setting because it does not rely on additional predicted masks or foundation models but solely uses the predicted object masks from the proposed model. For both evaluation protocols, we used mean Intersection over Union (mIoU) as an evaluation metric, which is widely used to measure segmentation performance. Additionally, we utilized the harmonic mean of the results from the seen and unseen categories as the final evaluation metric.

**Implementation Details.** We build upon CLIPSeg [32, 46], a CLIP-based encoder-decoder model. The implementation details can be found in the supplementary material.

### 4.2 Performance Evaluation

**Zero-Shot Part Segmentation.** We compare our PartCLIPSeg to previous methods [11, 32, 46, 52] on three OVPS benchmarks [7, 57]. As shown in Table 1, PartCLIPSeg consistently outperforms previous approaches by significant margins on Pascal-Part-116, demonstrating its zero-shot ability, with performance improvements of 3.94% in the Pred-All setting and 3.55% in the Oracle-Obj setting. The more challenging ADE20K-Part-234 dataset, which is a fine-grained segmentation dataset, further highlights the effectiveness of PartCLIPSeg. As shown in Table 2, PartCLIPSeg achieves a harmonic mean mIoU of 11.38% in the Pred-All setting, outperforming the best-performing baseline by 7.85%. In the Oracle-Obj setting, it achieves 38.60%, which is 4.45% higher than the best baseline. Notably, PartCLIPSeg shows significant performance improvement in unseen categories, demonstrating its strong generalizability. Considering that performance in unseen categories is crucial in a zero-shot

Table 1: Comparison of zero-shot performance with state-of-the-art methods on Pascal-Part-116.

| Method | Backbone | Pred-All | | | Oracle-Obj | | |
|---|---|---|---|---|---|---|---|
| | | Seen | Unseen | Harmonic | Seen | Unseen | Harmonic |
| ZSSeg+ [52] | ResNet-50 | 38.05 | 3.38 | 6.20 | **54.43** | 19.04 | 28.21 |
| VLPart [40] | ResNet-50 | 35.21 | 9.04 | 14.39 | 42.61 | 18.70 | 25.99 |
| CLIPSeg [32, 46] | ViT-B/16 | 27.79 | 13.27 | 17.96 | 48.91 | 27.54 | 35.24 |
| CAT-Seg [11, 46] | ViT-B/16 | 28.17 | **25.42** | **26.72** | 36.20 | **28.72** | 32.03 |
| PartCLIPSeg (Ours) | ViT-B/16 | **43.91**$_{\pm0.45}$ | 23.56$_{\pm0.21}$ | **30.67**$_{\pm0.09}$ (+3.94) | 50.02$_{\pm0.51}$ | **31.67**$_{\pm0.29}$ | **38.79**$_{\pm0.13}$ (+3.55) |

[1] The best score is **bold** and the second-best score is underlined. The standard error of an average of 5 results is reported. These are the same for all experiments.

Table 2: Comparison of zero-shot performance with state-of-the-art methods on ADE20K-Part-234.

| Method | Backbone | Pred-All | | | Oracle-Obj | | |
|---|---|---|---|---|---|---|---|
| | | Seen | Unseen | Harmonic | Seen | Unseen | Harmonic |
| ZSSeg+ [52] | ResNet-50 | **32.20** | 0.89 | 1.74 | **43.19** | 27.84 | 33.85 |
| CLIPSeg [32, 46] | ViT-B/16 | 3.14 | 0.55 | 0.93 | 38.15 | 30.92 | 34.15 |
| CAT-Seg [11, 46] | ViT-B/16 | 7.02 | 2.36 | 3.53 | 33.80 | 25.93 | 29.34 |
| PartCLIPSeg (Ours) | ViT-B/16 | 14.15$_{\pm0.51}$ | **9.52**$_{\pm0.13}$ | **11.38**$_{\pm0.10}$ (+7.85) | 38.37$_{\pm0.14}$ | **38.82**$_{\pm0.31}$ | **38.60**$_{\pm0.08}$ (+4.45) |

Table 3: Comparison of zero-shot performance with state-of-the-art method on PartImageNet.

| Method | Backbone | Pred-All | | | Oracle-Obj | | |
|---|---|---|---|---|---|---|---|
| | | Seen | Unseen | Harmonic | Seen | Unseen | Harmonic |
| CLIPSeg [32, 46] | ViT-B/16 | 32.39 | 12.27 | 17.80 | 53.91 | 37.17 | 44.00 |
| PartCLIPSeg (Ours) | ViT-B/16 | **38.82**$_{\pm0.74}$ | **19.47**$_{\pm0.45}$ | **25.94**$_{\pm0.32}$ (+8.14) | **56.26**$_{\pm0.29}$ | **51.65**$_{\pm0.62}$ | **53.85**$_{\pm0.37}$ (+9.85) |

scenario, these results are significant despite some performance degradation in seen categories. We also evaluated PartCLIPSeg on PartImageNet. According to Table 3, PartCLIPSeg shows a notable improvement over CLIPSeg.

We present the segmentation results of PartCLIPSeg in comparison to state-of-the-art open-vocabulary part segmentation methods [11, 32, 40] on Pascal-Part-116. Specifically, we focus on **qualitative performance** on unseen categories such as "dog", "sheep", "car", and "bird". As shown in Figure 5 for the Pred-All and Figure 6 for the Oracle-Obj setting, the proposed method effectively segments target parts regardless of the need for predefined masks during inference. Notably, PartCLIPSeg excels at identifying smaller, often overlooked part classes such as "eye", "tail", and "headlight". Additionally, our method effectively segments multiple objects and their respective parts, a challenge for other methods, demonstrating the effectiveness of PartCLIPSeg in zero-shot part segmentation. Its improved performance on unseen categories and higher accuracy in challenging environments highlight the robustness and generalization capabilities of PartCLIPSeg. Consistent improvements on Pascal-Part-116, ADE20K-Part-234, and PartImageNet demonstrate that PartCLIPSeg sets a new standard in open-vocabulary part segmentation.

**Cross-Dataset Part Segmentation.**

Table 4 validated the efficacy of our approach in a cross-dataset setting, where category names, annotation style, and granularity of mask may vary. Additionally, unlike zero-shot situations within the same dataset, there are differences in the types and diversity of parts. Initially, we trained our model on PartImageNet and ADE20K-Part-234 respectively. Subsequent tests on Pascal-Part-116 [7, 46] showed that PartCLIPSeg outperforms CLIPSeg in both the Pred-All and Oracle-Obj settings, confirming our method's superiority on generalization in different datasets.

Table 4: Cross-dataset performance.

| Method | Pred-All | Oracle-Obj |
|---|---|---|
| PartImageNet → Pascal-Part-116 | | |
| CLIPSeg [32, 46] | 11.72 | 14.87 |
| PartCLIPSeg (Ours) | **14.74** (+3.02) | **19.86** (+4.99) |
| ADE20K-Part-234 → Pascal-Part-116 | | |
| CLIPSeg [32, 46] | 5.41 | 17.82 |
| PartCLIPSeg (Ours) | **10.37** (+4.96) | **17.94** (+0.12) |

Table 5: Impact of attention control losses.

| $\mathcal{L}_{sep}$ | $\mathcal{L}_{enh}$ | Pred-All | | | Oracle-Obj | | |
|---|---|---|---|---|---|---|---|
| | | Seen | Unseen | Harmonic | Seen | Unseen | Harmonic |
| | | Pascal-Part-116 | | | | | |
| ✗ | ✗ | 43.86 | 21.89 | 29.20 | 49.09 | 31.26 | 38.20 |
| ✔ | ✗ | **44.01** | 23.18 | 30.37 | **50.37** | 31.45 | 38.72 |
| ✔ | ✔ | 43.91 | 23.56 | 30.67 | 50.02 | 31.67 | 38.79 |
| | | ADE20K-Part-234 | | | | | |
| ✗ | ✗ | 10.86 | 8.33 | 9.43 | 37.39 | 36.49 | 36.93 |
| ✔ | ✗ | 12.78 | 9.38 | 10.82 | 39.46 | 36.04 | 37.67 |
| ✔ | ✔ | 14.15 | 9.52 | 11.38 | 38.37 | 38.82 | 38.60 |

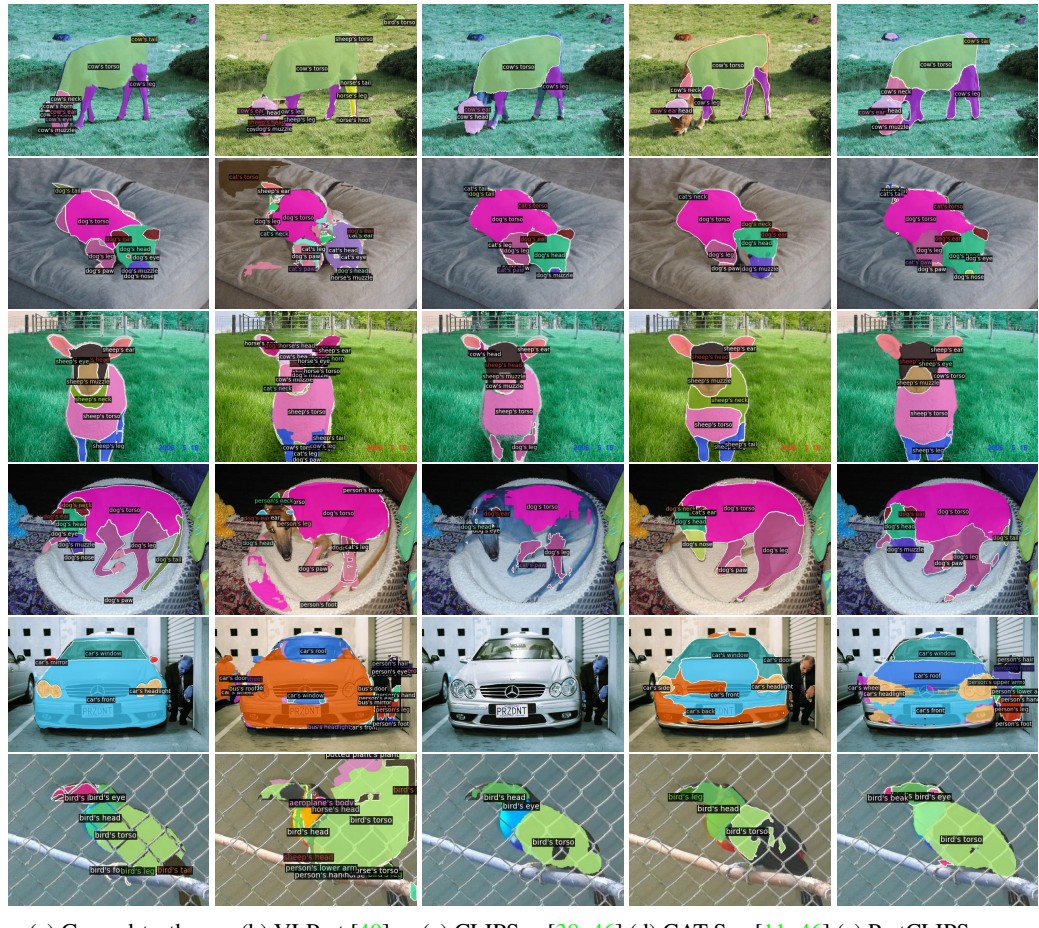

| (a) Ground-truth | (b) VLPart [40] | (c) CLIPSeg [38, 46] | (d) CAT-Seg [11, 46] | (e) PartCLIPSeg (Ours) |

Figure 5: Qualitative results of zero-shot part segmentation on Pascal-Part-116 in **Pred-All** setting. Annotations for unseen categories (bird, car, dog, sheep, etc.) are not included in the train set.

Table 6: Performance on mean Boundary IoU (↑) on Pascal-Part-116 in Oracle-Obj setting.

| Method | Seen | Unseen | Harmonic |
|---|---|---|---|
| ZSSeg+ [52] | 33.01 | 26.76 | 29.56 |
| CLIPSeg [32, 46] | 34.67 | 32.20 | 33.39 |
| CAT-Seg [11, 46] | 34.17 | 30.14 | 32.03 |
| PartCLIPSeg (Ours) | **36.15** | **39.07** | **37.55** |

Table 7: Impact of PartCLIPSeg for small parts on Pascal-Part-116 in Oracle-Obj setting. (mIoU)

| Part: "eye" | bird | cat | cow | dog | sheep | person |
|---|---|---|---|---|---|---|
| CLIPSeg [32, 46] | **3.33** | 18.77 | 3.65 | 16.05 | 0.00 | 15.30 |
| PartCLIPSeg (Ours) | 1.95 | **31.01** | **28.16** | **32.79** | **0.67** | **29.16** |
| **Part: "neck"** | bird | cat | cow | dog | sheep | person |
| CLIPSeg [32, 46] | 19.09 | 6.57 | 0.78 | 8.12 | 8.47 | 30.93 |
| PartCLIPSeg (Ours) | **32.51** | **12.00** | **2.75** | **16.37** | **18.80** | **50.71** |
| **Part: "leg"** | bird | cat | cow | dog | sheep | person |
| CLIPSeg [32, 46] | 19.61 | 38.62 | 27.85 | 39.34 | 52.63 | 52.67 |
| PartCLIPSeg (Ours) | **31.12** | **44.82** | **63.78** | **41.55** | **54.73** | **55.35** |

### 4.3 Ablation Study

In this section, we analyze the impact of each training loss on PartCLIPSeg. We focus on the roles of the separation and enhancement losses, examining how they contribute to improved segmentation accuracy.

**Separation & Enhancement Losses.** We conducted an ablation study to investigate the effect of the separation loss $\mathcal{L}_{sep}$ and the enhancement loss $\mathcal{L}_{enh}$ on the performance of PartCLIPSeg in Table 5. On Pascal-Part-116, eliminating both losses resulted in a lower harmonic mean of 29.20 in Pred-All and a harmonic mean of 38.20 in Oracle-Obj. Introducing $\mathcal{L}_{sep}$ without $\mathcal{L}_{enh}$ improved the harmonic mean in both Pred-All and Oracle-Obj setups. Using both losses led to the highest harmonic means of 30.67 and 38.79, respectively. Similarly, for ADE20K-Part-234, employing both losses resulted in the

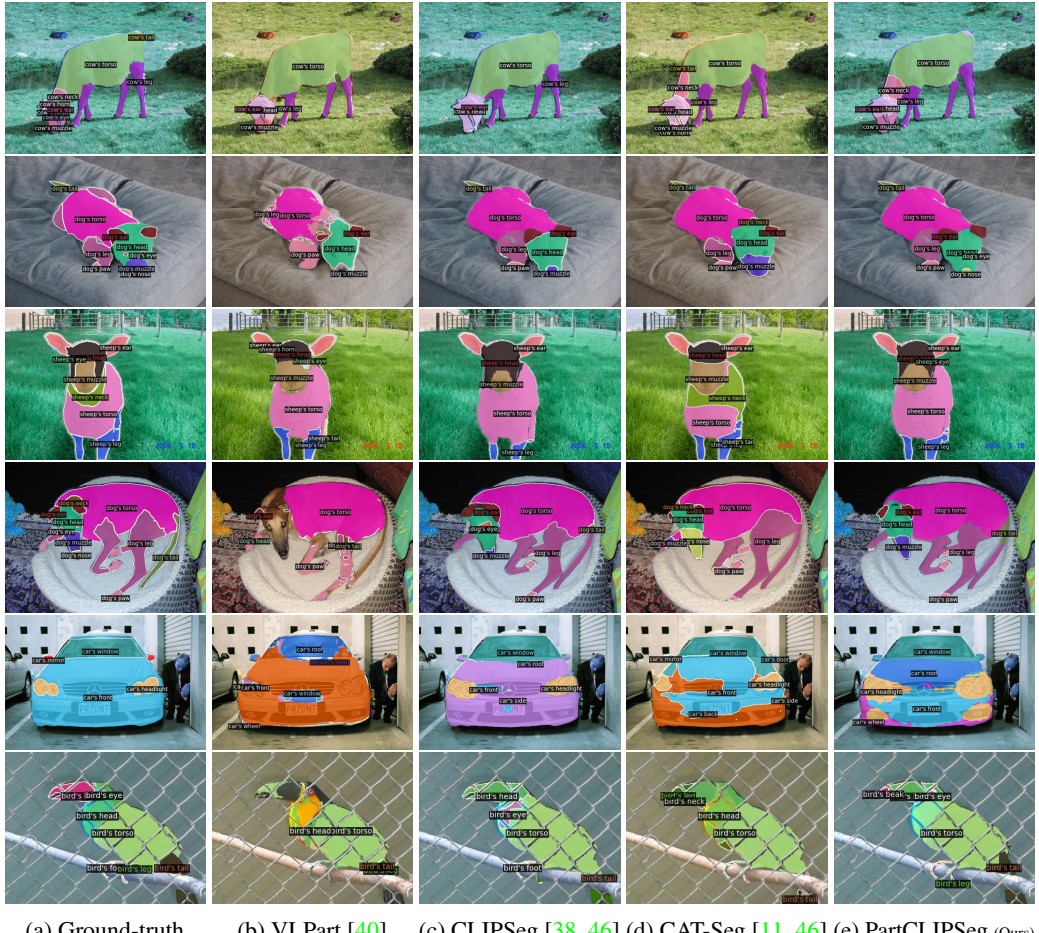

(a) Ground-truth     (b) VLPart [40]     (c) CLIPSeg [38, 46] (d) CAT-Seg [11, 46] (e) PartCLIPSeg (Ours)

Figure 6: Qualitative results of zero-shot part segmentation on Pascal-Part-116 in **Oracle-Obj** setting.

best performance, with harmonic means of 19.63 in Pred-All and 38.60 in Oracle-Obj. These results highlight the importance of both separation and enhancement losses in improving performance.

To verify the effectiveness of boundary creation of PartCLIPSeg, we examined an additional qualitative metric, Boundary IoU [8]. The results demonstrated high Boundary IoU performance, confirming that PartCLIPSeg effectively resolves ambiguous boundary issues as shown in Table 6.

**Impact of PartCLIPSeg for Underrepresented Parts.** We investigate the effect of the enhancement loss $\mathcal{L}_{enh}$ on OVPS model performance, especially with respect to underrepresented parts. In Table 7, we compare our PartCLIPSeg with CLIPSeg [32, 46] on small parts such as "eye", "neck", and "leg" of animals in Pascal-Part-116. As shown in the table, PartCLIPSeg consistently outperforms CLIPSeg with significant improvements in most cases. Notably, there is an impressive performance increase of 35.93%p for "cow's leg". These improvements highlight the effectiveness of the enhancement loss in accurately segmenting small and intricate parts, demonstrating its crucial role in improving overall performance.

## 5 Conclusion

In this study, we introduced PartCLIPSeg, a state-of-the-art OVPS method that addresses three primary challenges in OVPS. PartCLIPSeg utilizes generalized parts and object-level guidance to effectively solve identification issues. Then, it separates parts by minimizing their overlaps in attention maps, thus learning ambiguous part boundaries. Additionally, we implemented an enhancement loss function to improve the detection of underrepresented parts. Through extensive experimentation, we have confirmed the superior performance of PartCLIPSeg.

## Acknowledgements

This research was supported by the Basic Science Research Program through the National Research Foundation of Korea (NRF) funded by the MSIP (NRF-2022R1A2C3011154, RS-2023-00219019), KEIT grant funded by the Korean government (MOTIE) (No. 2022-0-00680, No. 2022-0-01045), the IITP grant funded by the Korean government (MSIT) (No. 2021-0-02068 Artificial Intelligence Innovation Hub, RS-2019-II190075 Artificial Intelligence Graduate School Program (KAIST)), and Samsung Electronics Co., Ltd (IO230508-06190-01).

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

# Supplementary Material

**Table of Contents**

# A   Discussion

## A.1   Limitations & Future Work

We share some limitations of our model and outline directions for future research. Our model is based on semantic segmentation, which does not allow for the discrimination of individual parts as instances. Consequently, parts such as "Paw 1" from *Dog 1* and "Paw 2" from *Dog 2* are assigned the same label. We plan to address this limitation in our future work to enhance the model's capability to distinguish between similar parts from different instances.

Furthermore, we believe that adding more inductive biases related to the relationships between parts, similar to key point detection which incorporates structural understanding, could yield higher-quality results.

Currently, our focus has been on object-specific parts, essentially mapping different granularity of vocabulary visually. Advanced methods could allow us to more effectively handle a broader variety of input categories, further enhancing our model's applicability and performance.

## A.2   Social Impact

This study explores open-vocabulary part segmentation, a technique that expands segmentation models to include fine-grained categories not encountered during training. The approach's robust nature allows for segmentation across various categories, proving invaluable for applications requiring flexibility and adaptability.

Open-vocabulary part segmentation could greatly influence several advanced fields. In robotics, for example, robots can precisely identify and handle a wide array of objects and components, essential for tasks from manufacturing assembly lines to complex medical surgeries. This adaptability allows robots to function in new settings without extensive retraining.

In healthcare, this technology enhances diagnostic processes by allowing for the segmentation of novel anatomical structures in medical imaging. This could facilitate earlier disease detection by identifying subtle, non-cataloged abnormalities essential for diagnosis.

In image editing, open-vocabulary part segmentation enables sophisticated manipulation by letting editors modify image fine-grained components not predefined in their software. This is especially

beneficial in the creative industries, where precise adjustments can improve output quality and foster innovation.

Adopting open-vocabulary part segmentation promises to enhance the efficiency, accessibility, and effectiveness of these technologies, particularly in handling real-world variability and unpredictability.

# B  Experimental Details

## B.1  Datasets Details

### B.1.1  Pascal-Part-116

In the Pascal-Part-116 dataset [7, 46], we target the following object-specific category names in Table A1. Among these, "bird", "car", "dog", "sheep", and "motorbike" are designated as unseen categories, encountered for the first time during inference in the zero-shot part segmentation setting.

Table A1: List of object-specific classes in Pascal-Part-116.

| Object-specific Part Categories | | | | |
| --- | --- | --- | --- | --- |
| aeroplane's body | aeroplane's stern | aeroplane's wing | aeroplane's tail | aeroplane's engine |
| aeroplane's wheel | bicycle's wheel | bicycle's saddle | bicycle's handlebar | bicycle's chainwheel |
| bicycle's headlight | bird's wing | bird's tail | bird's head | bird's eye |
| bird's beak | bird's torso | bird's neck | bird's leg | bird's foot |
| bottle's body | bottle's cap | bus's wheel | bus's headlight | bus's front |
| bus's side | bus's back | bus's roof | bus's mirror | bus's license plate |
| bus's door | bus's window | car's wheel | car's headlight | car's front |
| car's side | car's back | car's roof | car's mirror | car's license plate |
| car's door | car's window | cat's tail | cat's head | cat's eye |
| cat's torso | cat's neck | cat's leg | cat's nose | cat's paw |
| cat's ear | cow's tail | cow's head | cow's eye | cow's torso |
| cow's neck | cow's leg | cow's ear | cow's muzzle | cow's horn |
| dog's tail | dog's head | dog's eye | dog's torso | dog's neck |
| dog's leg | dog's nose | dog's paw | dog's ear | dog's muzzle |
| horse's tail | horse's head | horse's eye | horse's torso | horse's neck |
| horse's leg | horse's ear | horse's muzzle | horse's hoof | motorbike's wheel |
| motorbike's saddle | motorbike's handlebar | motorbike's headlight | person's head | person's eye |
| person's torso | person's neck | person's leg | person's foot | person's nose |
| person's ear | person's eyebrow | person's mouth | person's hair | person's lower arm |
| person's upper arm | person's hand | pottedplant's pot | pottedplant's plant | sheep's tail |
| sheep's head | sheep's eye | sheep's torso | sheep's neck | sheep's leg |
| sheep's ear | sheep's muzzle | sheep's horn | train's headlight | train's head |
| train's front | train's side | train's back | train's roof | train's coach |
| tvmonitor's screen | | | | |

### B.1.2  ADE20K-Part-234

In the ADE20K-Part-234 dataset [57], we target specific object categories listed in Table A2. The dataset includes 44 object classes and detailed subdivisions into over 200 part categories. Notably, "bench", "bus", "fan", "desk", "stool", "truck", "van", "swivel chair", "oven", "ottoman", and "kitchen island" are identified as novel classes and are encountered for the first time during inference in our zero-shot part segmentation setting.

### B.1.3  PartImageNet

PartImageNet [21] is a dataset derived from ImageNet [14], consisting of approximately 24,000 images across 158 classes. Each class has annotations for parts. All classes belong to one of 11 superclasses, organized using the hierarchical information from WordNet [34].

Previous open-vocabulary part segmentation research [40] primarily used PartImageNet to evaluate cross-dataset settings. In our study, we use PartImageNet not only for cross-dataset evaluation but also to assess model performance in zero-shot settings specific to PartImageNet.

To measure more generalized performance, we select 40 classes out of the 158. We maintain the proportion of existing superclasses as much as possible. For each superclass, at least 50% of the categories are designated as seen categories, with the remaining being unseen categories. Therefore, there are 25 seen classes and 15 unseen classes in our PartImageNet evaluation dataset.

Table A2: List of object-specific classes in ADE20K-Part-234.

| Object-specific Part Categories | | | | |
|---|---|---|---|---|
| person's arm | person's back | person's foot | person's gaze | person's hand |
| person's head | person's leg | person's neck | person's torso | door's door frame |
| door's handle | door's knob | door's panel | clock's face | clock's frame |
| toilet's bowl | toilet's cistern | toilet's lid | cabinet's door | cabinet's drawer |
| cabinet's front | cabinet's shelf | cabinet's side | cabinet's skirt | cabinet's top |
| sink's bowl | sink's faucet | sink's pedestal | sink's tap | sink's top |
| lamp's arm | lamp's base | lamp's canopy | lamp's column | lamp's cord |
| lamp's highlight | lamp's light source | lamp's shade | lamp's tube | sconce's arm |
| sconce's backplate | sconce's highlight | sconce's light source | sconce's shade | chair's apron |
| chair's arm | chair's back | chair's base | chair's leg | chair's seat |
| chair's seat cushion | chair's skirt | chair's stretcher | chest of drawers's apron | chest of drawers's door |
| chest of drawers's drawer | chest of drawers's front | chest of drawers's leg | chandelier's arm | chandelier's bulb |
| chandelier's canopy | chandelier's chain | chandelier's cord | chandelier's highlight | chandelier's light source |
| chandelier's shade | bed's footboard | bed's headboard | bed's leg | bed's side rail |
| table's apron | table's drawer | table's leg | table's shelf | table's top |
| table's wheel | armchair's apron | armchair's arm | armchair's back | armchair's back pillow |
| armchair's leg | armchair's seat | armchair's seat base | armchair's seat cushion | ottoman's back |
| ottoman's leg | ottoman's seat | shelf's door | shelf's drawer | shelf's front |
| shelf's shelf | swivel chair's back | swivel chair's base | swivel chair's seat | swivel chair's wheel |
| fan's blade | fan's canopy | fan's tube | coffee table's leg | coffee table's top |
| stool's leg | stool's seat | sofa's arm | sofa's back | sofa's back pillow |
| sofa's leg | sofa's seat base | sofa's seat cushion | sofa's skirt | computer's computer case |
| computer's keyboard | computer's monitor | computer's mouse | desk's apron | desk's door |
| desk's drawer | desk's leg | desk's shelf | desk's top | wardrobe's door |
| wardrobe's drawer | wardrobe's front | wardrobe's leg | wardrobe's mirror | wardrobe's top |
| car's bumper | car's door | car's headlight | car's hood | car's license plate |
| car's logo | car's mirror | car's wheel | car's window | car's wiper |
| bus's bumper | bus's door | bus's headlight | bus's license plate | bus's logo |
| bus's mirror | bus's wheel | bus's window | bus's wiper | oven's button panel |
| oven's door | oven's drawer | oven's top | cooking stove's burner | cooking stove's button panel |
| cooking stove's door | cooking stove's drawer | cooking stove's oven | cooking stove's stove | microwave's button panel |
| microwave's door | microwave's front | microwave's side | microwave's top | microwave's window |
| refrigerator's button panel | refrigerator's door | refrigerator's drawer | refrigerator's side | kitchen island's door |
| kitchen island's drawer | kitchen island's front | kitchen island's side | kitchen island's top | dishwasher's button panel |
| dishwasher's handle | dishwasher's skirt | bookcase's door | bookcase's drawer | bookcase's front |
| bookcase's side | television receiver's base | television receiver's buttons | television receiver's frame | television receiver's keys |
| television receiver's screen | television receiver's speaker | glass's base | glass's bowl | glass's opening |
| glass's stem | pool table's bed | pool table's leg | pool table's pocket | van's bumper |
| van's door | van's headlight | van's license plate | van's logo | van's mirror |
| van's taillight | van's wheel | van's window | van's wiper | airplane's door |
| airplane's fuselage | airplane's landing gear | airplane's propeller | airplane's stabilizer | airplane's turbine engine |
| airplane's wing | truck's bumper | truck's door | truck's headlight | truck's license plate |
| truck's logo | truck's mirror | truck's wheel | truck's window | minibike's license plate |
| minibike's mirror | minibike's seat | minibike's wheel | washer's button panel | washer's door |
| washer's front | washer's side | bench's arm | bench's back | bench's leg |
| bench's seat | traffic light's housing | traffic light's pole | light's aperture | light's canopy |
| light's diffusor | light's highlight | light's light source | light's shade | |

We conduct the dataset evaluation as follows: Models are trained on a training dataset composed of seen classes. Segmentation performance are then assessed on a validation dataset containing both seen and unseen classes. Evaluations were conducted in both Pred-All and Oracle-Obj settings.

Table A3: List of selected object classes per superclass. We choose 40 object classes from 158 categories to evaluate performance on PartImageNet and in a cross-dataset setting. Object categories that are both underlined and in **bold** represent the unseen classes, which are emphasized for their unique characteristics within each superclass.

| Superclass | Object Categories |
|---|---|
| Quadruped | tiger, giant panda, leopard, gazelle, **ice bear**, **impala**, **golden retriever** |
| Snake | green mamba, **Indian cobra** |
| Reptile | green lizard, Komodo dragon, tree frog, **box turtle**, **American alligator** |
| Boat | yawl, pirate, **schooner** |
| Fish | barracouta, goldfish, killer whale, **tench** |
| Bird | albatross, goose, **bald eagle** |
| Car | garbage truck, minibus, ambulance, **jeep**, **school bus** |
| Bicycle | mountain bike, moped, **motor scooter** |
| Biped | gorilla, orangutan, **chimpanzee** |
| Bottle | beer bottle, water bottle, **wine bottle** |
| Aeroplane | warplane, **airliner** |

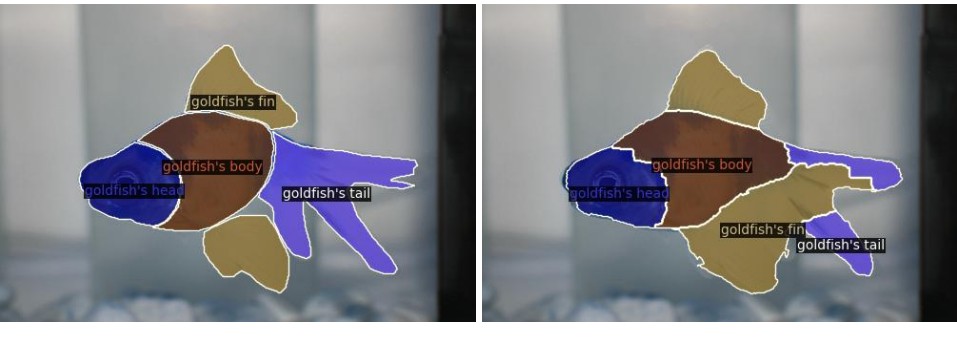

(a) Ground Truth        (b) Result from PartCLIPSeg (Oracle-Obj)

Figure A1: Example of part annotations in PartImageNet on our experiment.

## B.2  Implementation Details

Our model implementation is based on the CLIPSeg [32] architecture, as described in the OV-PARTS [46]. We utilized the pre-trained CLIP ViT-B/16 [38, 59] image encoder and text encoder for our experiments.

The model is trained using the ADAMW optimizer with a base learning rate of 0.0001 over 20,000 iterations, with a batch size of 8 images. We employ a WarmupPolyLR learning rate scheduler to manage the learning rate throughout the training process. To ensure model stability, we apply gradient clipping with a maximum gradient norm of 0.01.

We save model parameters every 1,000 iterations during training. The best-performing parameters are selected based on the highest validation evaluation scores. For example, the evaluation result on the Pascal-Part-116 dataset in the Oracle-Obj setting is derived from the checkpoint saved at the 5,000-step mark, which yields the best validation performance.

We evaluated several baseline methods—ZSSeg+, CLIPSeg [32], and CAT-Seg [11]—which are fine-tuned on our datasets. ZSSeg+ is a modified version of ZSseg [52], utilizing different fine-tuning methods according to [46]. It employs a ResNet-101 backbone and Compositional Prompt Tuning based on CoOp.

CLIPSeg and CAT-Seg models are pre-trained on object datasets; however, we fine-tuned these models on each part-level dataset. CAT-Seg, based on ResNet-101 and using ViT-B/16 as CLIP's visual encoder, achieved comparable performance by computing cost volumes and subsequently applying cost aggregation—a process that enhances segmentation by aggregating matching costs between image features. Specifically, CAT-Seg uses the frozen upsampling decoder but fine-tuned CLIP's image and text encoders. Conversely, we fine-tune the CLIPSeg decoder to better identify small segments and define clear boundaries. CLIPSeg, based on the ViT-B/16 architecture, is fine-tuned on the visual adapter, text embeddings, and transformer decoder to enhance its segmentation capabilities.

## B.3  Computational Resource

All our experiments are conducted on $8 \times$ NVIDIA A6000 GPUs.

As shown in Table A4, PartCLIPSeg offers advantages in both the number of parameters and memory consumption compared to other baselines on the Pascal-Part-116 dataset. With 152.4 million parameters, it is more efficient than ZSSeg+ and CAT-Seg, and comparable to CLIPSeg. In terms of GPU memory usage, PartCLIPSeg requires 24.4 GB, which is lower than both CAT-Seg and CLIPSeg.

Table A4: Computational resources on Pascal-Part-116 with batch size 8.

| Method | Params | Memory |
|---|---|---|
| ZSSeg+ [52] | 191.6 M | 11.1 G |
| CLIPSeg [32, 46] | 151.7 M | 25.5 G |
| CAT-Seg [11] | 180.6 M | 29.0 G |
| PartCLIPSeg | 152.4 M | 24.4 G |

For PartCLIPSeg, although the number of parameters is larger than CLIPSeg because of computations related to attention control, there is an advantage in not having to maintain weights for each object-

specific part due to the use of generalized parts. These efficiencies become more pronounced as the number of generalized parts shared among object classes increases. By leveraging shared representations for generalized parts, PartCLIPSeg reduces redundancy and memory requirements. This makes our model particularly advantageous in datasets where object classes have many common parts, leading to more efficient training and inference without compromising performance.

## C   Additional Quantitative Evaluation

Table A5: Recall performance on Pascal-Part-116 under the Oracle-Obj setting.

| Method | Seen | Unseen | Harmonic |
|---|---|---|---|
| ZSSeg+ [52] | 65.47 | 32.13 | 43.10 |
| CLIPSeg [32, 46] | 55.71 | 43.35 | 48.76 |
| CAT-Seg [11] | 56.00 | 43.20 | 48.77 |
| PartCLIPSeg (w/o $\mathcal{L}_{\tt sep} + \mathcal{L}_{\tt enh}$) | **58.97** | 46.47 | 51.98 |
| PartCLIPSeg (w/ $\mathcal{L}_{\tt sep} + \mathcal{L}_{\tt enh}$) | 58.46 | **47.93** | **52.67** |

Table A6: Recall performance on ADE20K-Part-234 under the Oracle-Obj setting.

| Method | Seen | Unseen | Harmonic |
|---|---|---|---|
| ZSSeg+ [52] | 55.78 | 40.71 | 47.07 |
| CLIPSeg [32, 46] | 49.59 | 48.11 | 48.84 |
| CAT-Seg [11] | 43.48 | 39.87 | 41.60 |
| PartCLIPSeg (w/o $\mathcal{L}_{\tt sep} + \mathcal{L}_{\tt enh}$) | 51.64 | 50.99 | 51.31 |
| PartCLIPSeg (w/ $\mathcal{L}_{\tt sep} + \mathcal{L}_{\tt enh}$) | **53.31** | **51.52** | **52.40** |

In this section, we present an additional evaluation metric that focuses on specific challenges within the Open-Vocabulary Part Segmentation (OVPS) task as shown in Figure 2. The Recall metric is used to assess how well the model captures underrepresented parts, addressing the challenge of underrepresented parts. Higher values in recall indicate that the model effectively captures these seldom-occurring parts, thereby addressing the challenge of underrepresented parts in OVPS.

PartCLIPSeg consistently achieves higher recall on both seen and unseen classes across both datasets as shown in Tables A5 and A6. The improved harmonic mean indicates that our model is more effective at identifying underrepresented parts, thereby addressing one of the core challenges in OVPS.

We further analyze the impact of the attention control losses $\mathcal{L}_{\tt sep}$ and $\mathcal{L}_{\tt enh}$ on the recall. By comparing the recall metric with and without these losses, we assess their effectiveness in enhancing the representation of seldom-occurring parts. From Tables A5 and A6, we observe that incorporating the attention control losses enhances the model's performance on unseen classes, which often include underrepresented parts. The increases in harmonic mean suggest that the attention control losses help the model to better capture these seldom-occurring or small parts.

## D   Additional Ablation

### D.1   Impact of Object-Level and Part-Level Guidance

Table A7: Ablation on $\lambda_{\tt obj}$, $\lambda_{\tt part}$, and attention control on Pascal-Part-116 in Oracle-Obj setting.

| $\lambda_{\tt obj}$ | $\lambda_{\tt part}$ | $\mathcal{L}_{\tt sep} + \mathcal{L}_{\tt enh}$ | Seen | Unseen | Harmonic mIoU |
|---|---|---|---|---|---|
| 0.0 | 0.0 | ✔ | 48.36 | 29.42 | 36.58 |
| 1.0 | 0.0 | ✔ | 48.61 | 31.28 | 38.07 |
| 0.0 | 1.0 | ✔ | 48.94 | **31.68** | 38.46 |
| 1.0 | 1.0 | ✘ | 49.09 | 31.26 | 38.20 |
| 1.0 | 1.0 | ✔ | **50.02** | 31.67 | **38.79** |

We conduct additional experiments to verify the impact of object-level and part-level label guidance on model performance as shown in Table A7. Specifically, we vary the weights $\lambda_{\tt obj}$ and $\lambda_{\tt part}$ in Equation (5), setting each to 0 or 1, to assess the influence of object-level and part-level supervision on the overall performance. Additionally, we evaluate the effect of the attention control losses, $\mathcal{L}_{\tt sep}$ and $\mathcal{L}_{\tt enh}$, by including or excluding them.

As shown in Table A7, both object-level and part-level guidance positively impact model performance on the Pascal-Part-116 dataset under the Oracle-Obj setting. When neither object-level nor part-level supervision is applied, the harmonic mean is 36.58. Introducing object-level guidance alone increases the harmonic mean IoU to 38.07, while part-level guidance alone raises it to 38.46. Combining both guidances yields the best performance with a harmonic mean IoU of 38.79.

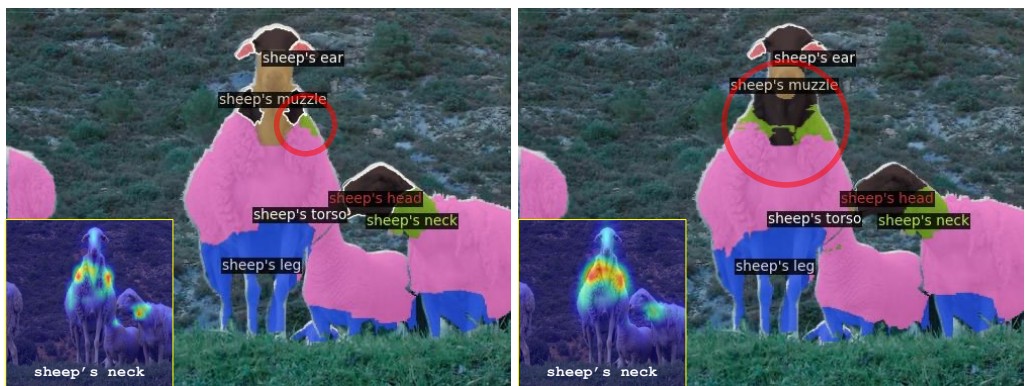

Figure A2: Comparison of results using only $\mathcal{L}_{\text{sep}}$ (top) with both $\mathcal{L}_{\text{sep}}$ and $\mathcal{L}_{\text{enh}}$ (bottom). The heatmap illustrates attention activation for the "sheep's neck" class.

Table A8: Effect of varying threshold $\gamma$ on Pascal-Part-116 in Oracle-Obj setting.

| Threshold ($\gamma$) | Seen | Unseen | Harmonic mIoU |
|---|---|---|---|
| 0.1 | 47.34 | **32.24** | 38.35 |
| 0.2 | 47.45 | 32.20 | 38.37 |
| 0.3 | **50.02** | 31.67 | **38.79** |
| 0.4 | 51.10 | 31.18 | 38.73 |
| 0.5 | 48.71 | 31.16 | 38.01 |

Additionally, removing the attention control losses $\mathcal{L}_{\text{sep}}$ and $\mathcal{L}_{\text{enh}}$ while keeping both $\lambda_{\text{obj}}$ and $\lambda_{\text{part}}$ at 1.0 results in a lower Harmonic mean of 38.20. This indicates that the attention control losses contribute to better distinguishing between seen and unseen classes.

## D.2 Qualitative Ablation on Attention Control Losses

The separation loss reduces the overlap between different parts, while the enhancement loss strengthens the activation of underrepresented parts. As shown in Figure A2, when only the separation loss $\mathcal{L}_{\text{sep}}$ is applied (top), smaller parts adjacent to larger parts may be diminished. Specifically, "sheep's neck" is not properly highlighted because minimizing the intersection can cause larger parts, such as the "sheep's torso" and "sheep's head", to overshadow smaller ones. When both losses $\mathcal{L}_{\text{sep}}$ and $\mathcal{L}_{\text{enh}}$ are utilized (bottom), the model accurately segments the small part—"sheep's neck"—as the enhancement loss boosts its representation, preventing it from being overwhelmed by larger neighboring parts.

This demonstrates that the separation and enhancement losses complement each other. Their combined use is essential to effectively distinguish and represent both large and small parts within an object, leading to improved segmentation performance.

## D.3 Ablation on the Hyperparameter in Attention Control

To evaluate the sensitivity of our method to the hyperparameter threshold $\gamma$ in Equation (8), we conducted experiments on the Pascal-Part-116 dataset under the Oracle-Obj setting. We varied $\gamma$ from 0.1 to 0.5 and measured the performance in terms of mIoU for seen and unseen classes, as well as the harmonic mean.

As shown in Table A8, our method is robust to the choice of $\gamma$ within the range of 0.1 to 0.5. The harmonic mean remains relatively stable, with the best performance achieved at $\gamma = 0.3$. While there is a slight variation in performance across different values of $\gamma$, the changes are not significant, indicating that our method does not heavily depend on the exact value of this hyperparameter.

# E    Additional Qualitative Results and Qualitative Analysis

## E.1    CLIP Embedding

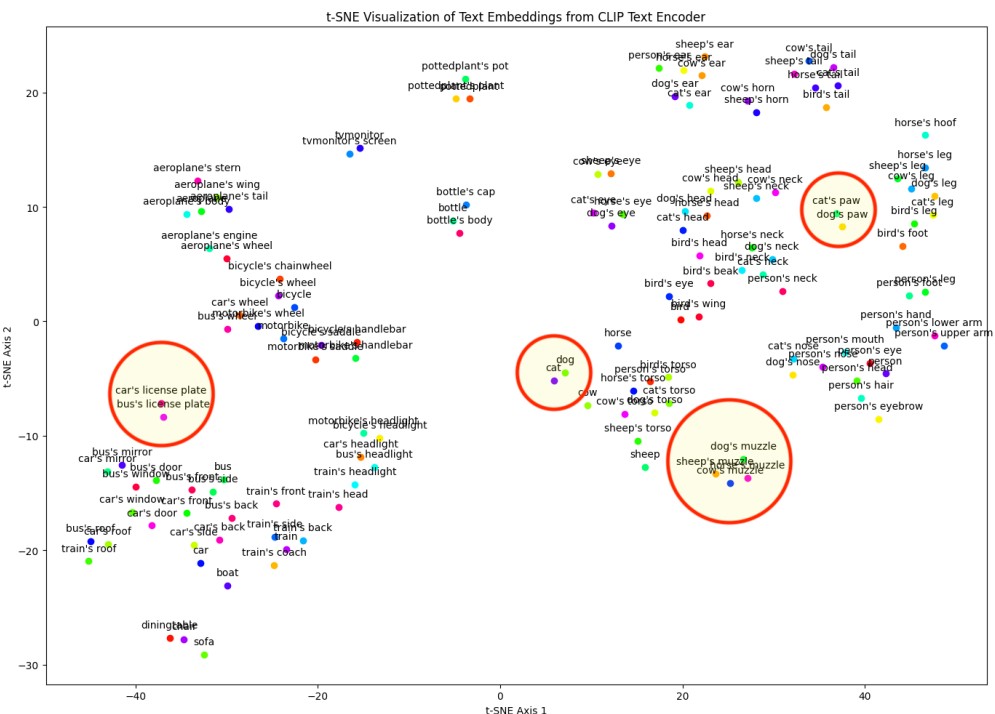

Figure A3: The t-SNE visualization of text embeddings from a pre-trained CLIP model on the classes of the Pascal-Part-116 dataset.

The t-SNE visualization of text embeddings from a pre-trained CLIP [38, 32, 58] model on the Pascal-Part-116 dataset [7, 46] reveals intriguing insights into the model's understanding of categories. Notably, similar classes such as "cats" and "dogs" are clustered closely within the embedding space. This proximity indicates a shared semantic space for categories that are visually or contextually related.

Additionally, we observed that object-specific parts sharing generalized parts, such as "car's license plate" and "bus's license plate", are also positioned near each other. This clustering suggests that the CLIP recognizes and leverages common parts across different objects that share common characteristics. Further analysis shows that object-specific classes containing parts like "muzzle" and "paw" are distributed in similar regions of the space. This consistency across different object categories emphasizes the CLIP's ability to generalize part-level features effectively.

Leveraging CLIP's text embeddings provides a significant zero-shot capability in the visual domain. This capability can be extended to part-level categories, demonstrating the potential for sophisticated unsupervised or zero-shot learning approaches in fine-grained object and part recognition tasks.

## E.2 Additional Qualitative Results

### E.2.1 Oracle-Obj Setting

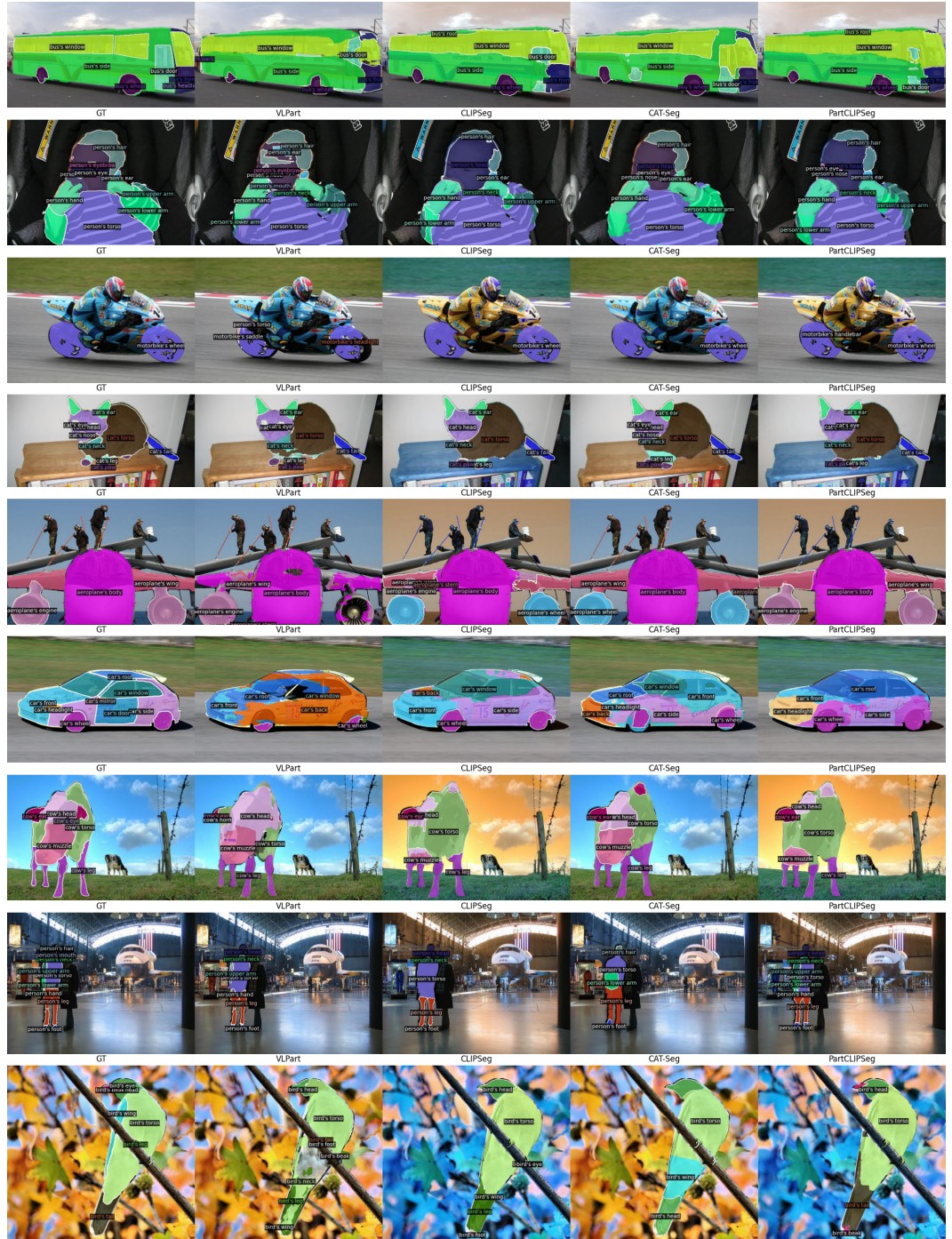

Figure A4: Comparison of VLPart, CLIPSeg, CAT-Seg, and our model on the Pascal-Part-116 dataset in Oracle-Obj setting.

## E.2.2 Pred-All Setting

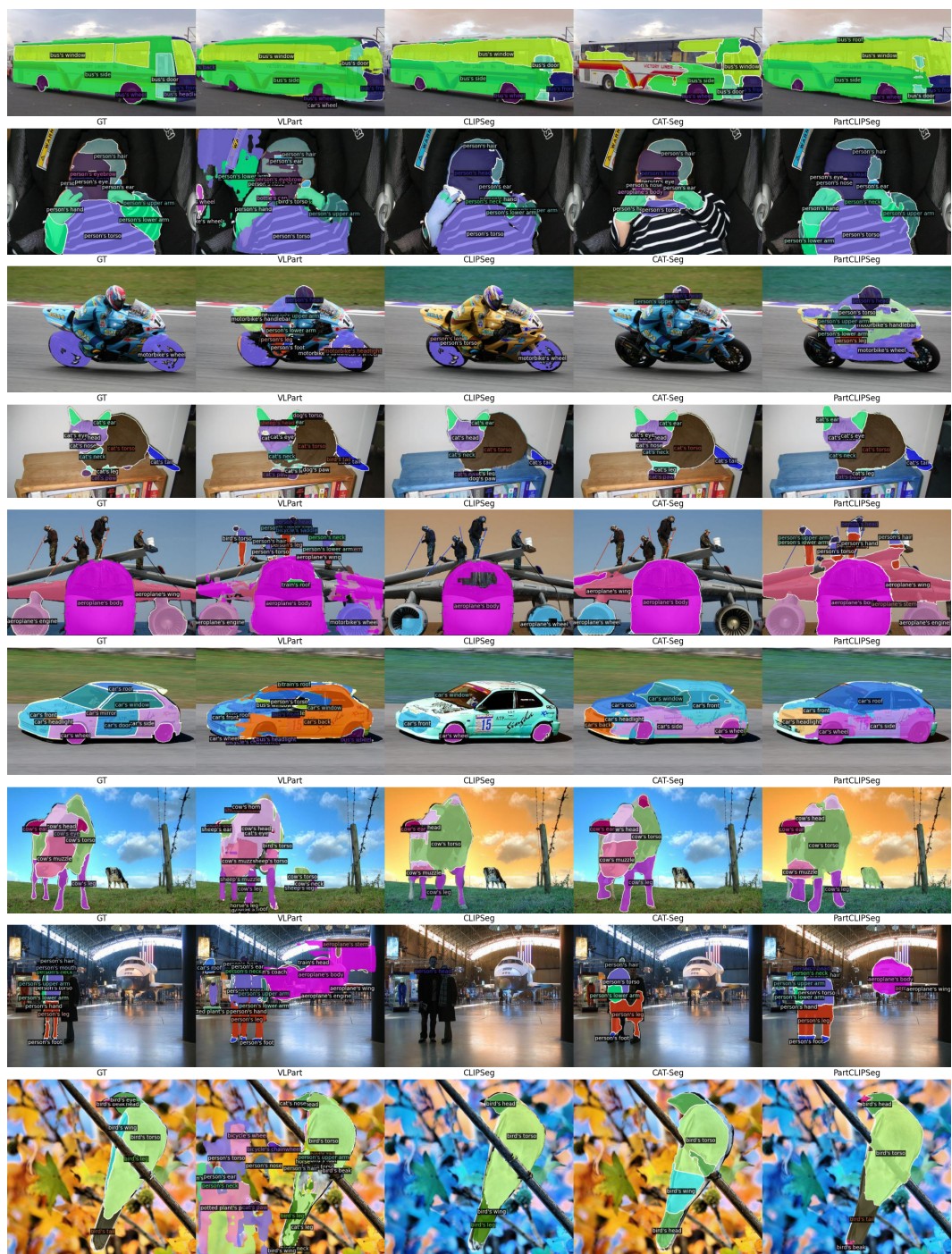

Figure A5: Comparison of VLPart, CLIPSeg, CAT-Seg, and our model on the Pascal-Part-116 dataset in Pred-All setting.

