# OpenReview forum: "Understanding Multi-Granularity for Open-Vocabulary Part Segmentation"
_NeurIPS.cc/2024/Conference — NeurIPS 2024 poster_

### Official Review · Reviewer_Vaxb · 2024-07-11

**Soundness:** 3
**Presentation:** 3
**Contribution:** 3
**Rating:** 6
**Confidence:** 4

**Summary:**

This paper proposed PartCLIPSeg for open-vocabulary part segmentation. This framework leverages generalized parts and object-level contexts for generalization in fine-grained parts, integrating competitive part relationships and attention control techniques. Extensive experiments on three datasets demonstrate the superior performance of PartCLIPSeg.

**Strengths:**

1) The paper is well-organized, good writing and easy to understand.
2) The topic of open-vocabulary part segmentation is meaningful and underexplored.
3) Significant improvement compared to the baseline methods.

**Weaknesses:**

1) Figure 3 (overall architecture) can be refined to clearly emphasize the contribution.
2) The analysis and explanation of the experiment are not sufficient. Although the method achieves a significant improvement on unseen categories, the results of seen categories on the Pred-All setting of the three datasets show different trends. Specifically, the results of the seen categories show a decrease compared to the baseline in Tables 1 and 3 (There exists an annotation mistake (41.70).), and a significant improvement in Table 2. Please discuss this result.
3) Ablations of loss functions. We notice that adding some loss function in some cases may lead to suboptimal results and show different trends on different datasets. For example, the second column of the Pred-All setting in Table 5. Please discuss this difference.

**Questions:**

How is the threshold γ in Equation 7 determined? How does this threshold affect the results?

**Limitations:**

The unseen categories of the three datasets in the experiment may have been seen during the CLIP pre-training, and we want to explore how well PartCLIPSeg generalizes to the “truly” unseen categories that have not been seen in the pre-training or from other fields

---

> ### Author Rebuttal · Authors · 2024-08-07
>
> Thank you for the detailed review and valuable suggestions.
>
> ## **Weakness 1**: Fig. 3 Refinement
>
> We refined Fig. 3 to make it clearer and to effectively emphasize our contribution.
> Specifically, we had made modifications to distinguish between the modules involved in training, fine-tuning, and the frozen.
> Additionally, we have delineated the data and operational components and added dotted lines to expand the explanation, highlighting our contribution more clearly.
>
> ## **Weakness 2**: Discussion in Experiment Results
>
>
> Thank you for pointing out the annotation mistake. We have corrected it.
>
> PartCLIPSeg enhances generalization ability by disentangling object-specific part guidance into object-level context and generalized parts. This improves performance for unseen classes compared to previous methods. However, this focus on generalization may cause some performance degradation for seen classes.
>
> Previous approaches often overfit to seen classes, resulting in high performance on seen classes but low performance on unseen classes.
> On the other hand, our approach leverages generalized information extracted from supervised seen classes, which can sometimes reduce performance on seen classes.
>
> Different trends depending on datasets: In ADE20K-Part-234, which contains many diverse and small objects (as shown in Fig. R5), the use of object-level priors enhances generalization performance.
> This is because directly predicting parts smaller than the small objects is much more challenging.
> However, in Pred-All setup of Pascal-Part-116 and PartImageNet, can lead to sub-optimal outcomes compared to directly using only object-specific part information if the object-level prior is noisy.
>
> Nonetheless, this framework allows us to leverage advanced methods for acquiring the object level prior, as demonstrated by gains in the Oracle-Obj setting.
> Specifically, modern approaches like SAM [C1], BEiT [C2], and EVA [C3], which excel in object-level predictions can support the validity of our approach.
> Since these models improve the accuracy of object-level prior, they can mitigate the performance degradation in the seen classes.
>
> Future research on OVPS can explore methods to maintain performance on seen categories while improving performance on unseen categories.
>
>
> [C1] Kirillov, Alexander, et al. "Segment anything." ICCV 2023.
>
> [C2] Bao, Hangbo, et al. "Beit: Bert pre-training of image transformers." ICLR 2022
>
> [C3] Fang, Yuxin, et al. "Eva: Exploring the limits of masked visual representation learning at scale." CVPR 2023.
>
>
>
>
>
>
> ## **Question 1**: Threshold γ
>
> The threshold was set empirically, and we have included the evaluation results for other values (0.1, 0.2, …, 0.5) as in the below table. It shows that our method is robust to the choice of threshold. We will add more detailed information about gamma in the main text and supplementary materials to provide further clarity.
>
> (Pascal-Part-116 / Oracle-Obj / mIoU)
>
> |threshold (γ)|seen|unseen|harmonic mean|
> |:-:|:-:|:-:|:-:|
> |.1|47.34|32.24|38.35|
> |.2|47.45|32.20|38.37|
> |.3|50.02|31.67|38.79|
> |.4|51.10|31.18|38.73|
> |.5|48.71|31.16|38.01|
>
> ## **Limitation**: Generalization to Truly Unseen
>
> PartCLIPSeg leverages the results of CLIP's pretraining, which, as you noted, presents a limitation in generalizing to categories not seen during CLIP's pretraining. This is a common limitation shared by methods across various open-vocabulary tasks, not just Part Segmentation. We agree that further research is needed to address the challenge of "truly" unseen categories in the future.
>
> The potential future work to address the challenge of truly unseen categories could explore the utilization of one-shot training and visual correspondence methods. One-shot training offers the potential to extend the model's capabilities to new categories with minimal training examples. By incorporating visual correspondence, there is a possibility to enhance the model's adaptability and improve its performance in recognizing and segmenting unseen categories. This approach could mitigate the limitations of pretraining and enable more flexible and effective handling of truly unseen scenarios.

---

> > ### Author Response · Authors · 2024-08-08
> >
> > We apologize for the oversight in our latest rebuttal. The response to Weakness 3 was accidentally omitted. We regret any confusion this may have caused.
> >
> > The response to Weakness 3 is provided below.
> >
> > ----
> >
> > ## **Weakness 3**: Loss Functions Ablations Discussion
> >
> > We will provide additional explanations to the initial explanation to address missing details.
> >
> > The separation loss reduces the overlap between parts, while the enhancement loss strengthens the underrepresented parts. These two losses are not independent; true effectiveness can be achieved when used together.
> >
> > Without the enhancement loss, the separation loss can lead to smaller parts being diminished when adjacent to larger parts. As illustrated in Fig. R2, applying only the separation loss results in missing small parts: minimizing the intersection may cause larger parts, such as sheep’s torso, head, and muzzle, to overshadow smaller parts, like sheep’s neck. Therefore, the enhancement loss is essential to ensure that the small parts, in the bottom row of Fig. R2, are accurately segmented and not overwhelmed by larger neighboring parts.
> >
> > We hope this additional explanation clarifies the role of separation and enhancement losses.

---

> > ### Comment · Area_Chair_XBW8 · 2024-08-12
> >
> > Hi dear reviewer,
> >
> > Please read through rebuttal, and see if the rebuttal has addressed your concerns or you have further comments.
> >
> > Thanks,
> >
> > AC

---

> > > ### Comment · Reviewer_Vaxb · 2024-08-12
> > >
> > > After revisiting the original paper, reviewing the comments from other reviewers, and considering the authors' responses, I find that my raised issues have been addressed. I believe the research topic of open-vocabulary part segmentation is meaningful, and this paper provides valuable research and very good results. I recommend accepting this paper. At this point, I tend towards keeping my original score.

---

> > > > ### Author Response · Authors · 2024-08-13
> > > >
> > > > We appreciate your thorough review and your acknowledgment of the significance of our research on open-vocabulary part segmentation. We are glad that the concerns raised have been fully resolved and are thankful for your recommendation to accept the paper.

---

### Official Review · Reviewer_LDit · 2024-07-12

**Soundness:** 2
**Presentation:** 3
**Contribution:** 2
**Rating:** 5
**Confidence:** 4

**Summary:**

This paper proposes PartCLIPSeg, which builds upon the CLIPSeg model and extends it to handle the unique challenges of part segmentation. This paper introduces several components to address the challenges in OVPS.

1. Generalized parts with object-level contexts: This approach combines generalized part information with object-level context to mitigate the lack of generalization in fine-grained parts.

2. Attention control: This method introduces competitive part relationships and attention control techniques to minimize overlap between predicted parts and enhance activation for underrepresented parts. This mechanism is a new addition to the field.

The paper conducts experiments on OVPS’s baseline datasets. The result shows that PartCLIPSeg outperforms existing SOTA methods in most settings. The Ablation primarily focuses on the impact of the proposed attention control loss, which lacks a comprehensive analysis of other key components of this paper.

**Strengths:**

- Introducing generalized parts with object-level contexts: The paper proposes an approach that combines generalized part information with object-level context to enhance the model’s ability to generalize to unseen object-part combinations.
- Developing an attention control mechanism: The paper introduces an attention control mechanism which helps to address the issues of ambiguous part boundaries and missing small or infrequent parts.
- Promising result on baseline benchmark: The paper demonstrates state-of-the-art performance in most settings of OVPS.

**Weaknesses:**

- Insufficient ablation studies: The ablation studies primarily focus on the proposed attention control loss, which is not sufficient to prove the effectiveness of the proposed method.
     1) The impact of combining object-level and part-level pseudo-label is not explored.
     2) The paper does not provide sufficient evidence that the proposed losses work as intended, beyond their impact on the final metric. Further discussion is needed to validate whether the losses effectively address the challenges they were designed to tackle.
- Limited comparison with other OVPS methods: The paper compares PartCLIPSeg primarily with CLIPSeg and CATSeg, which are designed for open-vocabulary segmentation and not specifically for OVPS. The comparison with other OVPS methods is lacking.
- Limited discussion on computational complexity: Since the paper introduces a new attention control mechanism, a detailed analysis of the computational requirements and speed trade off could be helpful for a more comprehensive understanding of the proposed method.
Inadequate qualitative analysis: Insufficient exploration and insights of the model’s performance on challenging and failure cases.

**Questions:**

See Weakness.

**Limitations:**

No societal impact.

---

> ### Author Rebuttal · Authors · 2024-08-07
>
> Thank you for your insightful comments and suggestions.
>
> ## **Weakness 1.1**: Impact of Object-Level and Part-Level Labels
>
>
> We conducted additional experiments to verify the impact of object-level and part-level pseudo-labels.
> We varied $\lambda_1$ and $\lambda_2$ in Equation 5, setting each to 0 and 1 in turn, to assess the influence of each guidance on the overall model performance.
>
> Table R2 shows that reducing the weight of the object-level loss ($\lambda_1$) improved performance in the Oracle-Obj setting (with provided object-level masks). On the other hand, increasing the weight of object-level loss enhanced performance in the Pred-All setting. This indicates the advantage of focusing on part information prediction when object-level masks are available (Oracle-Obj setting).
>
>
> ## **Weakness 1.2**: Validation of Addressing Challenges
>
> We initially integrated the solutions to the three challenges into the main body of the paper.
> However, we realized that this approach compromised clarity.
> Based on the precious feedback, we have included additional qualitative assessments to demonstrate how each issue has been resolved in Fig. R1 of the attached PDF.
>
> Also, we conducted additional quantitative studies on boundary IoU [C1] to validate the accuracy of the boundaries as shown in Table R3.
>
> Overall (in addition to Fig. R1)
>
> * (a) Lack of generalization:
>     * Fig. 5, row 3 shows improvements in detecting areas that models like CLIPSeg struggled with in the Pred-All setting, indicating enhanced generalization performance.
> * (b) Ambiguous boundaries:
>     * Fig. 5, row 1 illustrates how competitive part segmentation helps mitigate ambiguous boundaries.
>     * Additional metric mean boundary IoU (in Table R3) verified the superiority of PartCLIPSeg.
> * (c) Missing underrepresented parts:
>     * Fig. 5, row 2, and Fig. 4 highlight improvements in the segmentation of underrepresented parts.
>     * We also have detailed this in Section 4.3 through an ablation study.
>     * Fig. R4 demonstrates that our method helps identify small part classes.
>
>
> [C1] Cheng, Bowen, et al. "Boundary IoU: Improving object-centric image segmentation evaluation." CVPR 2021.
>
>
> ## **Weakness 2**: Comparison with Other OVPS Methods
>
> We have now included a performance comparison with VLPart (ICCV 2023) [C1].
>
> VLPart is a pioneering study in OVPS that uses a Mask R-CNN-based approach leveraging the DINO feature for correspondences.
>
> Notably, VLPart focuses on instance segmentation with masks. In contrast, our research compares semantic segmentation methods like ZSSeg, CATSeg, and CLIPSeg, following the recent OVPS research [C2]
> This distinction also led to the omission of a direct comparison with VLPart in the OpenReview of OV-PARTS (NeurIPS 2023) [C2].
>
> Based on the feedback, we recognize that a performance comparison would enhance the overall understanding and context.
> Therefore, we conducted experiments under the same conditions to compare VLPart with our method.
> We adapted the mask outputs of VLPart, which uses mAP-based evaluation, by converting them to the semantic segmentation label with the highest pixel-wise confidence.
>
> It's important to note that VLPart's original experiments on Pascal-Part only included dog and bus as unseen classes, while our current experimental setup on Pascal-Part-116 (following OV-PARTS) includes bird, car, dog, sheep, and motorbike. Therefore, we re-trained VLPart on Pascal-Part-116 to ensure a fair comparison with our method.
>
> As shown in Table R1, our results show that PartCLIPSeg outperforms VLPart in both the Pred-All setting and the Oracle-Obj setting.
> This is because VLPart relies on the nearest seen class when predicting unseen classes, whereas our model understands disentangled object-level and part-level contexts and utilizes attention control.
> We believe leveraging the nearest seen class can be disadvantageous when the number of unseen classes increases, especially if these unseen classes are unrelated or distant from the seen classes.
>
> The qualitative results can be found in Fig. R3.
>
> [C1] Sun, Peize, et al. "Going denser with open-vocabulary part segmentation.", ICCV 2023.
>
> [C2] Wei, Meng, et al. "OV-PARTS: Towards open-vocabulary part segmentation.", NeurIPS 2023.
>
>
> ## **Weakness 3**: Computational Requirements
>
> **[Computational complexity]** For PartCLIPSeg, although the parameters increase due to the computations related to attention control, there is an advantage in not having to maintain weights for each object-specific part due to the use of generalized parts. Detailed information regarding parameters and memory is provided in Table R4.
>
> **[Challenging and failure cases]** PartCLIPSeg uses ViT-based CLIP, and since the proposed attention modulation is dependent on the resolution of the attention map, it has limitations on achieving optimal performance for overlapping small parts.
>
>
> ## **Limitation**
>
> The section related to societal impact was added to the supplementary materials (Ln 30 - 47). Although this is still in its early stages and immediate applications are limited, we aim to present a foundation for future applications in areas such as autonomous robot guidance, image editing, and fine-grained understanding.

---

> > ### Comment · Area_Chair_XBW8 · 2024-08-12
> >
> > Hi dear reviewer,
> >
> > Please read through rebuttal, and see if the rebuttal has addressed your concerns or you have further comments.
> >
> > Thanks,
> >
> > AC

---

### Official Review · Reviewer_KLx2 · 2024-07-12

**Soundness:** 2
**Presentation:** 2
**Contribution:** 2
**Rating:** 5
**Confidence:** 4

**Summary:**

This paper identifies three key problems of current open-vocabulary part segmentation, namely, lack of generalization, ambiguous boundaries, and missing underrepresented parts. The paper then proceeds to propose solutions to fix these problems. Experimental results show that the proposed method outperforms previous state-of-the-art methods.

**Strengths:**

This paper identified some key issues in current open-vocabulary part segmentation and proposed effective solutions respectively. The paper was easy to follow, with good figures to demonstrate key concepts. The experiments also demonstrate that the methods can lead to nontrivial gains.

**Weaknesses:**

1. **Limited discussion of previous open-vocabulary part segmentation work and their relationship to this work.** For example, I quickly searched and read the VLPart paper [1], which seems to parse objects first, then parts, thus avoiding the object-level misclassification issue (or the "lack of generalization" as named by the authors). Indeed, the VLPart seems very natural to me. So I am actually surprised to find that there is such an issue (mixing objects and parts during classification) as indicated by this paper. In general, the related work section provides very limited introduction of the previous open-vocabulary part segmentation methods, esp. how they are related to the proposed approach. There are also **no descriptions of the compared baselines in the experiment section**.This made readers hard to make a fair assessment of the contribution of this paper.

2. **Limited ablation studies to understand the proposed approach.** Some key questions are unanswered: how much does each proposed component (object-level context, attention control) contribute to the performance gain? In what way or what types of data? It might also be helpful to compare the Pred-Obj performance because that helps disentangle the localization and classification ability of the approaches. Current comparison mixes many components together and it's difficult to gain insights about how the gain is achieved.

3. **Lack of addressing directly the identified key problems**. The paper starts with three key issues and then propose different ways to fix them. However, in the experiments, there are no direct addressing to these problems any more (except Table 6 for small parts). It would be great if the experiments can clearly demonstrate how the three problems are mitigated by the proposed approaches.

4. **Lack of discussion on the training details in the paper**. For readers who are not familiar with CLIPSeg, it is impossible to tell how the model is trained and what datasets are used for training. There is also **a lack of descriptions of the compared baselines in the paper**. Without these information, it's hard to understand the performance gains.

Minor issues:

1. I found it confusing to call it "lack of generalization" when the dog's parts are misclassified as sheep's or cat's. Isn't it just misclassification, or object-level misclassification?

2. Texts on the figures are way too small, esp. in Figure 2 and Figure 4.

3. The "supplementary material" for this paper should be put as an appendix to the main paper file, instead of as a separate file.

**Questions:**

1. What is "incomplete guidance from knowledge-based, multi-granularity characteristics of parts"? It is unclear from the texts. It seems like an important motivation for the design of the attention control. Some more analysis or discussion would help here.

2. In Table 1 and Table 3, why are the performance of PartCLIPSeg on "seen" classes so low? From the model design, I don't see a reason for this. Do the authors have an explanation?

3. How to understand the "failures" in Table 6 where PartCLIPSeg cannot improve the performance from the CLIPSeg (such as "bird's eye") or PartCLIPSeg also has very low performances (such as "sheep's eye", "cow's neck")? And are all these parts indeed *small*? For example, "cow's leg" doesn't seem to be small to me. How do the authors define what are small parts?

**Limitations:**

The authors have addressed the limitation and potential societal impact of the work.

---

> ### Author Rebuttal · Authors · 2024-08-07
>
> Thank you for your insightful comments and suggestions.
>
> ## **Weakness 1**: Discussion of Previous OVPS
>
> We have now included a performance comparison with VLPart (ICCV 2023).
>
> VLPart is a pioneering study in Open-vocabulary Part Segmentation (OVPS).
> VLPart focuses on instance segmentation with masks, whereas our research compares semantic segmentation methods like ZSSeg, CATSeg, and CLIPSeg.
> This distinction also led to the omission of a direct comparison in the OpenReview of OV-PARTS (NeurIPS 2023).
>
> However, based on the feedback, we recognize that a performance comparison would enhance the overall delivery of context.
> Therefore, we conducted experiments under the same conditions to compare VLPart with our method.
>
> Our results show that PartCLIPSeg outperforms VLPart in both the Pred-All and the Oracle-Obj settings.
> VLPart is based on a Mask R-CNN model; it may encounter errors at the object-level classification (in the Pred-All setting).
> Even if it detects parts within an object, the overall performance can be compromised if the object-level detection itself is not accurate.
> We have included the qualitative and quantitative evaluation results in Table R1 and Fig. R3.
>
> (Due to space limitations with other questions, we kindly ask you to refer to the response to Reviewer LDit's Weakness 2 for additional information.)
>
> ## **Weakness 2**: Ablation Studies
>
> Additional ablation studies related to object-level context and attention control were conducted.
> The experiments include various object-level and part-level guidance with and without attention control.
> This allowed us to analyze the influence of object-level guidance, part-level guidance, and attention loss through their respective coefficients.
>
> Detailed information is provided in Table R2.
>
> (Due to space limitations with other questions, we kindly ask you to refer to the response to Reviewer LDit's Weakness 1.1 for additional information.)
>
> ## **Weakness 3**: Addressing of Key Problems
>
> Based on the precious feedback, we have included additional qualitative assessments to demonstrate how each issue has been resolved in Fig. R1.
>
> Also, we conducted additional quantitative studies on boundary IoU(Intersection of Union) as in Table R3.
>
> (Due to space limitations with other questions, we respectfully refer you to the response to Reviewer LDit’s Weakness 1.2 for detailed information.)
>
>
> ## **Weakness 4**: Training Details and Baseline Descriptions
>
> We add more explanations regarding our training details and CLIPSeg, as well as descriptions of the baseline methodologies: ZSSeg+ and CATSeg in the main text.
>
> CLIPSeg: Extends a frozen CLIP model with a transformer-based decoder. It operates with both text and image prompts, using skip connections from the CLIP encoder to the decoder and modulating the decoder's inputs using FiLM conditioning. Through this, CLIPSeg inherits CLIP’s zero-shot ability and can be fine-tuned to different downstream tasks.
>
> Our method utilizes these CLIPSeg pre-trained on the PhraseCut dataset and frozen CLIP. We fine-tuned the model respectively for each experiment using the training set of datasets.
>
> ## **Minor issues**
>
> Regarding incorrect object-level prediction, it can indeed be seen as misclassification.
> However, we refer to it as a “lack of generalization” because it indicates an insufficient disentanglement of object-level and part-level information.
>
> Moreover, the misclassification in the unseen class suggests that the model does not fully generalize the holistic understanding and contextual information from the seen classes to the unseen classes. We will clarify this in the revised version.
>
> Thank you. We made the corrections for minor issues 2 and 3.
>
> ## **Question 1**: Characteristics of Parts
>
> Our intention was to convey that, unlike object-level segments, parts may not have clear, well-defined boundaries that may be defined by human convention or knowledge, making perfect guidance challenging.
> For instance, the "head" may include the neck or only the face.
> We highlight that supervision can be inherently ambiguous since the boundaries of parts are subjective and competitive, based on agreed terminology.
>
> This challenge motivates the design of our attention control mechanism.
> Instead of relying solely on ambiguous supervision, we use common knowledge that parts should not overlap significantly.
> Our approach focuses on reducing the overlap between parts while simultaneously enhancing underrepresented parts. We have further clarified this in the main text.
>
>
> ## **Question 2**: Performance on "Seen" Classes
>
> PartCLIPSeg is designed to improve generalization by focusing both on object-level context and generalized parts, which significantly enhances performance on unseen classes. However, the emphasis on generalization can sometimes lead to sub-optimal performance in seen classes.
>
> (Due to space limitations with other questions, we respectfully refer you to the response to Reviewer Vaxb's Weakness 2 for detailed information.)
>
> ## **Question 3**: Understanding Failures in Small Parts
>
> We define a small part as any part that occupies less than 20% of the object mask. We have added this definition for clarity.
>
> Our model, along with other OVPS methods, faces challenges in accurately predicting small parts. Specifically, for unseen classes such as "bird's eye," the proposed model still exhibits lower prediction accuracy due to the limited resolution of the attention map.
>
> We have included detailed information on the proportion of object-specific parts, like the “cow’s leg”, relative to the object-level mask in Figure R4. This provides additional context on the size of these parts in relation to the entire object.

---

> ### Comment · Reviewer_KLx2 · 2024-08-11
> **Thanks for the response and a few more comments**
>
> I first want to thank the detailed responses from the authors, esp. the added experiments, they are very helpful to understand the method. And I still have a few more comments:
>
> 1. Related to W1: "VLPart focuses on instance segmentation with masks, whereas our research compares semantic segmentation methods". Does it make sense to comparing semantic segmentation for this work in the first place? As far as I understand, this paper is targeting the part segmentation problem, then it definitely should treat it as at least an instance segmentation problem, or more appropriately, a panoptic segmentation problem. Especially given prior works VLPart already works on instance segmentation problem, I cannot understand the choice from the authors. I don't want to jump to negative guesses, so maybe the authors can give some more explanation on the chosen baselines and the reasoning behind it.
>
> 2. Related to W3: it's great to have more qualitative examples and additional quantitative studies on boundary IoU. But I still think it doesn't demonstrate clearly enough how the proposed method solves the key problems and why the method is able to. It's probably more about the paper presentation. At this point, it might even be helpful to rethink what's the key problems that the method has resolved and how to present them better. Then additional quantitative results would be useful to support the reasoning.
>
> 3. Final comments: I appreciate the authors adding many details like related works and training details. However, these should definitely have been present in the paper in the first place. These made me feel the paper was a bit hurried. To be frank, I am also a bit worried about whether a full comparison with other works is complete. While I did mention VLPart, that's because I only searched this paper. As a non-expert in this subfield, I am not sure whether there are many more or not. The lack of many details made me less confident of the overall paper. I will maintain my rating for now.

---

> > ### Author Response · Authors · 2024-08-12
> >
> > Thank you for providing the opportunity to clarify several points.
> >
> > ## **1. Related to W1**
> >
> > * VLPart (ICCV 2023) approaches OVPS through instance segmentation, while OV-PARTS (NeurIPS 2023) utilizes semantic segmentation. We followed the OV-PARTS protocol, which is the more recent and higher-performing method, due to the following reasons.
> >     * We believe that the lack of text-image alignment is a significant issue in the current OVPS landscape.
> >     * Addressing semantic segmentation first allows us to focus on the core alignment issues within OVPS.
> >     * OV-PARTS has demonstrated the most notable performance among existing OVPS methodologies, and we have used it as a baseline for our task.
> > * In response to the reviewer’s suggestion that instance segmentation may be more suitable for part segmentation, we believe that further research is needed, particularly in open-vocabulary settings.
> >     * While it may be logical for part segmentation to evolve into instance or panoptic segmentation in the long term, it is critical first to develop models that can extend to various categories under semantic segmentation, especially in zero-shot environments like open-vocabulary. For this reason, semantic segmentation is actively researched in object-level open-vocabulary domains (e.g. OVSS), as seen in recent works like CAT-Seg (CVPR 2024), PnP-OVSS (CVPR 2024), and FC-CLIP (NeurIPS 2023).
> >     * **Methods based on instance segmentation have not yet demonstrated effective performance in fine-grained recognition tasks like part segmentation compared to semantic segmentation**.
> >         * As shown in Table 4 of VLPart, despite being tested in a simpler experimental setup (with only two novel classes, dog and bus), VLPart exhibits low performance, with a mAP of 4.2 and mAP_50 of 11.0 on novel classes. Similar trends were also confirmed under our experimental conditions, as shown in Table R1.
> >         * These results highlight the limitations of instance-based segmentation, which do not outweigh its purported advantages.

---

> > ### Author Response · Authors · 2024-08-12
> >
> > ## **2. Related to W3**
> >
> > In this study, we identified three key challenges associated with existing OVPS methodologies (VLPart, OV-PARTS) and proposed an enhanced OVPS approach to address these issues: (a) Lack of generalization, (b) Ambiguous boundaries, and (c) Missing underrepresented parts. In the following, we clarify how the proposed approach addresses the key challenges.
> >
> > ##### **(a) Lack of generalization:**
> > To address generalization issues, we develop generalized part guidance and object guidance. (Section 3.2: L117-176)
> > Generalized part guidance is designed to learn the part class across (object-specific part) base classes in the training set so as to improve performance on novel classes in testing. Object guidance is used to resolve the ambiguity of the same part category name belonging to a different object class by using object-level supervision. Our results confirm that misclassification has reduced: qualitatively in Fig. 5 (row 3), Fig. R1 (a), Section 3.2.2 (suppl.), and quantitatively in Tables 1, 2, 3 (especially under the Pred-All setup), Table 4 (cross dataset), and Table R2 based on the m-IoU metric. An ablation study in Table R2 clearly justifies the roles of both guidances.
> >
> > ##### **(b) Ambiguous boundaries:**
> > To overcome the ambiguity in part-level supervision, we leverage the common understanding that these parts should be distinct and non-overlapping. To achieve this, we introduce a constraint $\mathcal{L}\_{sep}$, which ensures that attention activations from different object-specific parts remain disjoint within an object. (Section 3.3.1: L194-203).
> > It is supported by qualitative results in Fig. 5 (row 1) where ambiguous boundaries are mitigated, Fig. R1(b), and Section 3.2.2 (suppl.), where non-fully covered part predictions occur in other baselines, particularly in the Pred-All setting. Quantitative results are presented in Table R3, based on the boundary IoU metric. An ablation study in Tables A5 and A6 further demonstrates the effectiveness of our proposed loss function in mitigating ambiguous boundaries.
> >
> > ##### **(c) Missing underrepresented parts:**
> > To ensure that underrepresented parts, such as small or less frequent ones, are not ignored, we propose $ \mathcal{L}\_{enh} $. This is achieved by maximizing the attention activation values for these parts, allowing them to be more effectively identified. (Section 3.3.2: L204-214)
> > Our method is supported by qualitative results shown in Fig. 4, Fig. 5 (row 2), Fig. R1 (c), and Section 3.2.1 (suppl.) where small part classes such as bird’s tail and beak are correctly segmented. Quantitative results are provided in Table 6, Fig. R4, Table A1, and Table A2, using the metrics m-IoU and recall. Ablation studies on m-IoU in Section 4.3, Table 5, and Recall in Table A3, and Table A4 further highlight the effectiveness of our proposed method.
> >
> >
> > **(Dataset / Evaluation Setting / Metric)**
> >
> > **[Table A1] (Pascal-Part-116 / Oracle-Obj / Recall)**
> > |model|seen|unseen|harmonic Recall|
> > |:-:|:-:|:-:|:-:|
> > |ZSSeg+|65.47|32.13|43.10|
> > |CAT-Seg|56.00|43.20|48.77|
> > |CLIPSeg|55.71|43.35|48.76|
> > |PartCLIPSeg|58.46|47.93|52.67|
> >
> >
> > **[Table A2] (ADE20K-Part-234 / Oracle-Obj / Recall)**
> > |model|seen|unseen|harmonic Recall|
> > |:-:|:-:|:-:|:-:|
> > |ZSSeg+|55.78|40.71|47.07|
> > |CAT-Seg|43.48|39.87|41.60|
> > |CLIPSeg|49.59|48.11|48.84|
> > |PartCLIPSeg|53.31|51.52|52.40|
> >
> >
> > **[Table A3] (Pascal-Part-116 / Oracle-Obj / Recall)**
> > |model|seen|unseen|harmonic Recall|
> > |:-:|:-:|:-:|:-:|
> > |w/o $ \mathcal{L}\_{sep} + \mathcal{L}\_{enh} $ |58.97|46.47|51.98|
> > |w/ $ \mathcal{L}\_{sep} + \mathcal{L}\_{enh} $|58.46|47.93|52.67|
> >
> >
> > **[Table A4] (ADE20K-Part-234 / Oracle-Obj / Recall)**
> > |model|seen|unseen|harmonic Recall|
> > |:-:|:-:|:-:|:-:|
> > |w/o $ \mathcal{L}\_{sep} + \mathcal{L}\_{enh} $ |51.64|50.99|51.31|
> > |w/ $ \mathcal{L}\_{sep} + \mathcal{L}\_{enh} $|53.31|51.52|52.40|
> >
> >
> > **[Table A5] (Pascal-Part-116 / Oracle-Obj / B-IoU)**
> > |model|seen|unseen|harmonic B-IoU|
> > |:-:|:-:|:-:|:-:|
> > |w/o $ \mathcal{L}\_{sep} + \mathcal{L}\_{enh} $ |36.24|37.87|37.04|
> > |w/ $ \mathcal{L}\_{sep} + \mathcal{L}\_{enh} $|36.15|39.07|37.55|
> >
> >
> > **[Table A6] (ADE20K-Part-234 / Oracle-Obj / B-IoU)**
> > |model|seen|unseen|harmonic B-IoU|
> > |:-:|:-:|:-:|:-:|
> > |w/o $ \mathcal{L}\_{sep} + \mathcal{L}\_{enh} $ |24.99|22.41|23.63|
> > |w/ $ \mathcal{L}\_{sep} + \mathcal{L}\_{enh} $|25.67|22.46|23.96|

---

> ### Author Response · Authors · 2024-08-12
>
> ## **3. Final comments**
>
> Thank you for your feedback. We understand your concerns. However, we had a thorough understanding of the existing research (including VLPart) and excluded VLPart from the experimental comparison only due to differences in experimental protocols. We also emphasize that our experiments on semantic segmentation protocols are thorough and more competitive in terms of performance. Besides, we even suggest a new evaluation criterion, the Pred-All setting.
>
> * We focused on the semantic segmentation task of OVPS for the reasons mentioned earlier in answer 1.
>     * We adopted the state-of-the-art OVPS baseline from OV-PARTS and conducted a thorough comparison with all comparable methodologies in this task.
>     * Although VLPart was mentioned in the introduction and related work, a detailed comparison was initially omitted because VLPart adheres to the instance segmentation protocol. However, this has been strengthened thanks to the reviews (Table R1).
>     * Baselines such as ZSSeg+, CATSeg, and CLIPSeg (introduced in OV-PARTS) along with their training details were previously introduced in Section 2.2 of the supplementary materials under Implementation Details (L77-L98). However, based on the feedback, we will revise the related work to make these descriptions more prominent and ensure that they are not overlooked.
> * We were aware of existing OVPS studies, including VLPart, from the initial submission. OVPS is an emerging and underexplored research area with limited research focused on improving the direct alignment between images and part-level text.
>     * In the Part Segmentation section of the introduction and related work, we briefly described all the latest OVPS-related methods we could find at the time of submission, including VLPart [35] (L33-34, L84-86), OV-PARTS [40] (L34-37), and OPS [30] (L83-84).
>     * **To the best of our knowledge, OV-PARTS achieved the highest performance in OVPS and there have been no follow-up studies** specifically addressing the part-level semantic segmentation task in an OV setting without relying on additional mask proposal models, aside from our work.
>
>
> [30] OPS: Towards Open-World Segmentation of Parts (CVPR 2023)
>
> [35] VLPart: Going Denser with Open-Vocabulary Part Segmentation (ICCV 2023)
>
> [40] OV-PARTS: Towards Open-Vocabulary Part Segmentation (NeurIPS 2023 D&B)

---

> > ### Comment · Reviewer_KLx2 · 2024-08-12
> > **Thanks for the authors' responses**
> >
> > I want to thank the authors for their detailed responses and clarifications. They help clarify some of my misunderstandings. I re-read the paper and the other comments. Based on my final understanding, I decided to keep my current rating as borderline acceptance.

---

> > > ### Author Response · Authors · 2024-08-13
> > >
> > > We sincerely appreciate your thoughtful review and valuable comments.
> > > We are pleased to have resolved the questions raised and appreciate your positive decision.
> > > Thank you again for your time and insight.

---

### Author Rebuttal · Authors · 2024-08-07

Dear Reviewers,

We sincerely appreciate all reviewers for their thorough and insightful feedback.
The valuable reviews have significantly enhanced the overall delivery of the proposed method.

We are particularly thankful that all reviewers (Vaxb, LDit, KLx2) found our paper show promising results and non-trivial improvement.
We appreciate the reviewers for highlighting that our paper is well-organized and easy to follow (Vaxb, KLx2).
Additionally, we are grateful for noting that the task of OVPS is meaningful and underexplored (Vaxb).
Finally, we are thankful for acknowledging our identification of key issues in OVPS and our effective approach to addressing these issues (LDit, KLx2).

Through this review, we were able to answer the following questions. We kindly ask you to refer to the individual responses and the attached PDF.

The progress made based on the review can be summarized as follows.

* Additional Baseline: VLPart (ICCV 2023)
    * Additional experiments related to the OVPS baseline were conducted.
    * Qualitative and quantitative results of VLPart can be found in Table R1, Fig. R2.
* Qualitative and Quantitative Results of Initial Challenges
    * New qualitative results have been added in Fig. R1.
    * We conducted extra quantitative experiments of boundary IoU for Challenge B (Ambiguous boundaries), and results have been added in Table R3.
* Additional Ablation Studies and Analysis
    * Futher ablation studies have been conducted for different hyperparameters of $\lambda_1$, $\lambda_2$, attention control ($ {\mathcal{L}\_{sep}} $, $ {\mathcal{L}\_{enh}} $), and $ \gamma $ as in Table R2.
    * New qualitative explanations of attention losses are added in Fig. R2.
    * The analysis of the datasets in the aspect of object-specific part size has been added in Fig. R4 and the overall object size in Fig. R5.
    * Detailed computation resource information has been added in Table R4.


We refined the paper for clarity and hope our responses address the reviewers' concerns.
We are happy to answer any further questions.

We sincerely thank you again for the valuable feedback.

---

> ### Comment · Area_Chair_XBW8 · 2024-08-09
> **kicking off reviewer-author discussion**
>
> Thanks the authors for the rebuttal, and reviewers' for the comments. We are in the period of author-reviewer discussion.
> - Reviewers, please read through all the reviews and rebuttal and see if authors' responses have addressed your and others' important questions.
> - Authors, please follow up with any reviewers' further comments.
>
> Cheers,
>
> AC

---

### Decision · Program_Chairs · 2024-09-25

**Decision:**

Accept (poster)

**Comment:**

This is a borderline paper; three reviewers give positive ratings and and recommend to accept it. Reviewers think the module design is novel and the performance is good. However, reviewers are also concerned about that the paper (1) misses many related works and descriptions of training details, (2) fails to position in the literature of OV Part Seg to better motivate or compare the work, (3) has a large space to improve the writing quality. To prepare the camera-ready, authors should:
- include the missing citations,
- carefully re-write related paragraphs to better motivate the work and better position it in the literature,
- describe important implementation details
- polish the language and improve the readability of the paper.